# Genomic adaptation of giant viruses in polar oceans

**Lingjie Meng** [1], **Tom O. Delmont** [2,3], **Morgan Gaïa** [2,3], **Eric Pelletier** [2,3], **Antonio Fernàndez-Guerra** [4], **Samuel Chaffron** [3,5], **Russell Y. Neches** [1], **Junyi Wu** [1], **Hiroto Kaneko** [1], **Hisashi Endo** [1] & **Hiroyuki Ogata** [1] ✉

Despite being perennially frigid, polar oceans form an ecosystem hosting high and unique biodiversity. Various organisms show different adaptive strategies in this habitat, but how viruses adapt to this environment is largely unknown. Viruses of phyla *Nucleocytoviricota* and *Mirusviricota* are groups of eukaryote-infecting large and giant DNA viruses with genomes encoding a variety of functions. Here, by leveraging the Global Ocean Eukaryotic Viral database, we investigate the biogeography and functional repertoire of these viruses at a global scale. We first confirm the existence of an ecological barrier that clearly separates polar and nonpolar viral communities, and then demonstrate that temperature drives dramatic changes in the virus–host network at the polar–nonpolar boundary. Ancestral niche reconstruction suggests that adaptation of these viruses to polar conditions has occurred repeatedly over the course of evolution, with polar-adapted viruses in the modern ocean being scattered across their phylogeny. Numerous viral genes are specifically associated with polar adaptation, although most of their homologues are not identified as polar-adaptive genes in eukaryotes. These results suggest that giant viruses adapt to cold environments by changing their functional repertoire, and this viral evolutionary strategy is distinct from the polar adaptation strategy of their hosts.

Polar regions are recognized as among the coldest environments on Earth, with strong seasonal variations in light cycles. Nevertheless, high primary productivity of phytoplankton in these regions nourishes a diverse range of creatures, ranging from microscopic organisms to large animals. Organisms adapted to polar environments exhibit distinctive physiological or morphological characteristics that augment their fitness in these extreme environments: polar bears show characteristic morphological traits whose underlying genetic variations occurred in their ancestral gene pools;[1] Arctic and Antarctic fishes have evolved antifreeze proteins that allow them to maintain physiological activity in cold waters;[2,3] and some psychrophilic bacteria exhibit oxygen-scavenging enzymes or modify their membrane chemistry[4,5].

How do viruses adapt to polar environments? In the ocean, viruses are the most abundant biological entities[6] and play important roles in the regulation of microbial host communities, carbon and nutrient cycling, and horizontal gene transfer among organisms[7–10]. Some viruses are known to adapt to environments by acquiring specific metabolic genes. A notable example is cyanophages in low phosphorous environments, which tend to possess phosphorus assimilation genes[11]. Do viruses in polar environments have any specific genomic features? Recent metagenomic studies have revealed that

[1]Bioinformatics Center, Institute for Chemical Research, Kyoto University, Gokasho, Uji 611-0011, Japan. [2]Génomique Métabolique, Genoscope, Institut François Jacob, CEA, CNRS, Univ Evry, Université Paris-Saclay, F-91057 Evry, France. [3]Research Federation for the study of Global Ocean systems ecology and evolution, FR2022/Tara GOsee, F-75016 Paris, France. [4]Lundbeck Foundation GeoGenetics Centre, GLOBE Institute, University of Copenhagen, Copenhagen, Denmark. [5]Nantes Université, École Centrale Nantes, CNRS, LS2N, UMR 6004, F-44000 Nantes, France. ✉e-mail: ogata@kuicr.kyoto-u.ac.jp

both Arctic[12,13] and Antarctic[14,15] environments harbor diverse viruses, with an elevated diversity of prokaryotic dsDNA viruses in the Arctic Ocean[12]. A substantial proportion of genes specific to these viruses were suggested to be under positive selection based on the ratio of non-synonymous to synonymous mutation rates. While the function of most of these genes remains unknown, this result suggests that the gene repertoire plays a role in viral adaptation to Arctic regions[12]. Another study showed that a prokaryotic virus reduced its genome in response to decreased culture temperature[16]. It is also known that closely related viruses can display different responses in their infection dynamics to varying temperature[17,18], suggesting that temperature can select both viruses and their hosts. However, our knowledge on viruses in polar environments is still limited.

In our previous study, we revealed a remarkable shift in the community composition of eukaryotic dsDNA viruses from nonpolar to polar biomes[19]. These viruses, classified in phylum *Nucleocytoviricota* ("giant viruses"), are known for their large genomes encoding hundreds to thousands of genes[20,21]. These viruses are ancient[22], diverse[23,24], abundant[25,26], and active[27,28] in the ocean. Their genomes dynamically evolve by losing and gaining different functions[20,29]. Despite the existence of a clear polar/nonpolar barrier for these viruses, how frequently these viruses have crossed this polar barrier over evolutionary time remains unclear. Furthermore, how adaptation to polar environments impacted their gene repertoire is unknown.

In this study, we investigate the genomes of eukaryotic large DNA viruses to characterize the viral genome-level adaptation to polar environments. We leverage recently reconstructed viral and eukaryotic environmental genomes from the multidisciplinary *Tara* Oceans international research project[30,31]. The viral genomic data include environmental genomes of viruses from phylum *Nucleocytoviricota* and a recently discovered phylum, *Mirusviricota*[31]. Mirusviruses are large dsDNA viruses, which widely distribute in the global ocean, likely infect marine plankton, and share a large and similar gene repertoire with viruses of the *Nucleocytoviricota*.

We first assess the existence of a polar barrier for giant viruses by analyzing viral community composition and by computing robust temperature optima for viruses and their predicted hosts. We then perform ancestral state reconstruction for polar and nonpolar niches along the phylogenomic tree of these viruses to quantitatively estimate the adaptive evolutionary events. Finally, we delineate the functions that are specific to 'polar' viruses and present evidence that viral genomic adaptation to low temperature polar regions is distinct from the strategy of their hosts.

## Results and discussion
### Polar barrier for giant viruses
We investigated the biogeography of giant virus genomes from the Global Ocean Eukaryotic Viral (GOEV) database[31]. Their abundance profiles across *Tara* Oceans samples from different size fractions (Fig. 1a; Supplementary Fig. 1a; Supplementary Data 1) revealed 1380 viral genomes that showed signals (>25% of the genome length was mapped by reads, see methods in our previous paper[31]) in at least one sample out of 928 samples (The details of biogeography were in the supplementary text; Supplementary Figs. 1–3; Supplementary Data 2). The presence/absence distribution of viral genomes across biomes revealed a large number of genomes specific to the Polar biome. Out of 569 genomes detected in polar regions, 262 (46.05%) were exclusive to the Polar biome (Supplementary Fig. 4a). Accordingly, biome-based classification of viral communities (i.e., Polar, Coastal, Trades, and Westerlies) had significant explanatory power for community variation (Supplementary Fig. 4b,c; ANOSIM, $P < 0.01$). The $R$ value of ANOSIM test (intergroup dissimilarity)[32] increased from 0.4021 to 0.6141 after merging three nonpolar biomes, demonstrating the existence of a clear polar barrier for giant virus communities. The viral communities of Arctic regions were also characterized by their relatively high

abundances showing peaks in cumulative RPKM plots for different size fractions (Supplementary Fig. 1a). The major groups of viruses in this area were *Algavirales*, followed by *Imitervirales* as in other areas of the ocean (Supplementary Fig. 2c).

We inferred a virus–plankton network through co-occurrence analysis to further characterize the polar barrier in the context of virus-host interaction. In this analysis, we combined our virus genome data with previously reconstructed marine eukaryotic genome data[30]. In total, 2135 virus–eukaryote associations (edges) were inferred in the network, with the majority (91.94%) of them being positive associations (Fig. 1b; Supplementary Data 3). Virus–eukaryote pairs with strong associations (edge weight ≥0.4) showed significantly higher protein similarities between their genomes than those without strong associations (no edges or edges with weight <0.4) (Wilcoxon rank-sum test, $P = 1 \times 10^{-13}$) (Fig. 1c). Such an increase of sequence similarity can be due to horizontal gene transfers between viruses and hosts[33,34], supporting the prediction of true virus–host relationships in the reconstructed network. A previous study revealed that the structure of the network for marine eukaryotes and prokaryotes correlates with the temperature optima of species[35]. By estimating robust temperature optima for individual viruses and eukaryotes[36], we identified a strong correlation between the temperature optima and the structure of the virus–eukaryote network (Fig. 1b). A dramatic structural change in the network at the temperature-dependent polar/nonpolar boundary is the source of the uniqueness of polar viral communities.

Latitudinal diversity gradients are characterized by relatively low polar and high temperate biodiversity[37] and are widespread across all ranges of marine microorganisms[38]. Previous studies revealed a similar latitudinal diversity gradient for giant viruses[38], but not for prokaryotic dsDNA viruses[12,38] and RNA viruses[39]. In this study, various diversity gradient patterns were observed among viruses of different size fractions and main taxonomic groups (Supplementary Fig. 1c; Supplementary Fig. 5). We observed a mid-latitude peak in small-size fractions, and hotspots of viral diversity in the Arctic regions within large-size fractions. The reasons underlying the Arctic diversity hotspots for some viruses (e.g., viruses in large-size fractions and mirusviruses) may reflect their host ranges as previously suggested[38]. Notably, eukaryotic nodes (i.e., potential hosts) associated with viruses showed a pattern distinct from the general diversity gradient trend with increasing diversity towards high latitudinal regions (Supplementary Fig. 6).

### Potential hosts for polar viruses
We employed a phylogeny-informed filtering method, Taxon Interaction Mapper (TIM)[40,41], to map the virus-eukaryote edges of the network to the viral phylogenomic tree. This method links viruses and eukaryotes through a clade-to-clade relationship by testing whether leaves (i.e., viral genomes) under a node of the virus tree are enriched with a specific predicted host group. TIM assigned five predicted host taxa to 34 viral clades, covering 6.38% of total viral genomes ($N = 88$, Supplementary Fig. 7a). These predictions are summarized in Supplementary Fig. 7b and included known virus–host relationships: *Mesomimiviridae* (from *Imitervirales*) and Phaeocystales;[42–44] *Mesomimiviridae* and Pelagomonadales;[45,46] and *Prasinovirus* (from *Algavirales*) and Mamiellales[47,48].

Recent discoveries of giant endogenous viral elements (GEVEs) that are widespread across different eukaryotes demonstrated the impacts of giant viruses' infection on host genome evolution[9,34,49,50]. We first analyzed insertions of genomes of giant viruses and their satellite viruses (i.e., virophages) in marine eukaryotic genomes[30]. Among the five taxa of predicted viral hosts, the diatom order Chaetocerotales showed the largest number of insertion signals of both giant viruses and virophages (Supplementary Fig. 7b), suggesting infection of dsDNA viruses in Chaetocerotales. Genomes of two Chaetocerotales isolates also displayed a comparable level of

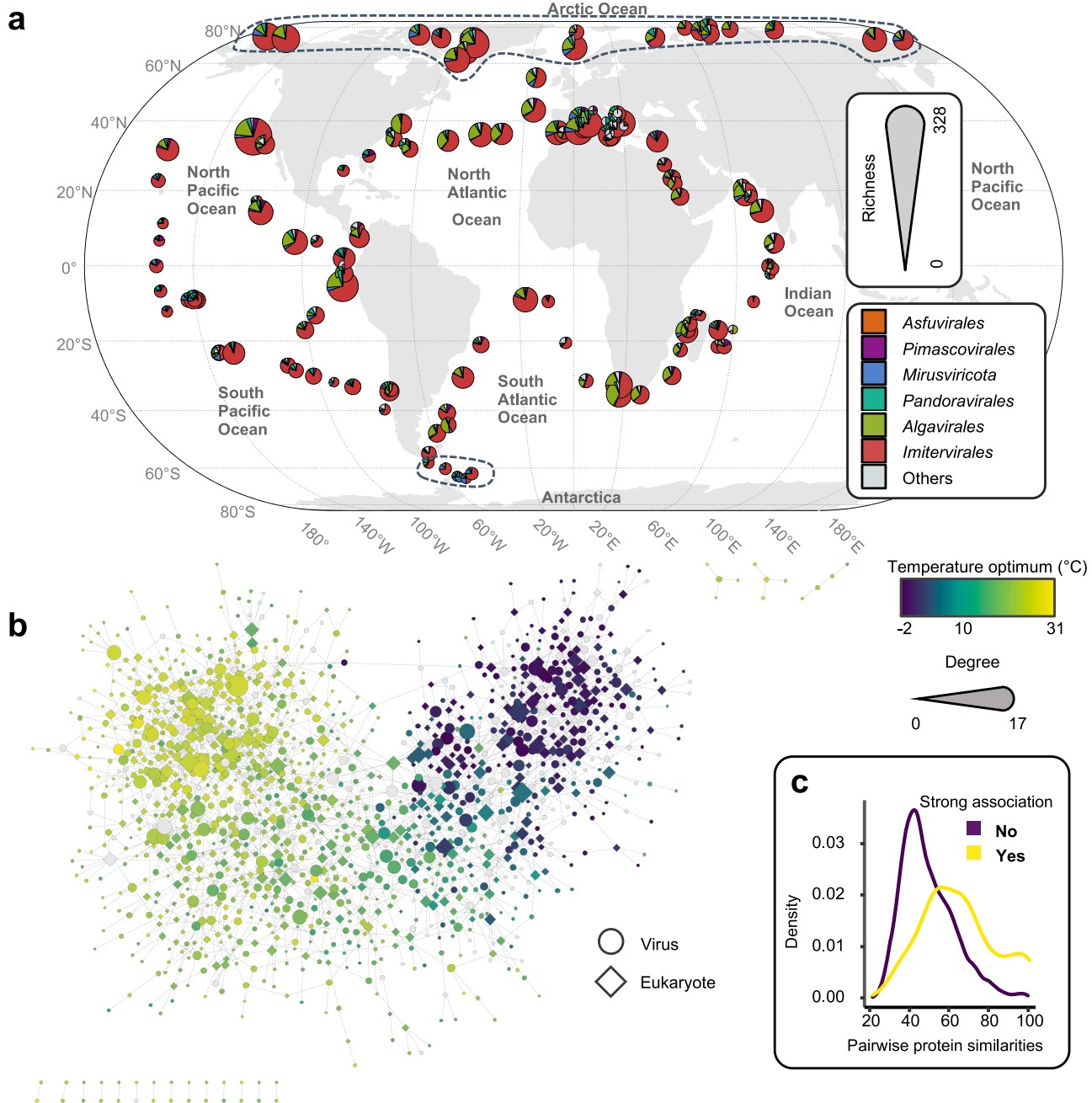

**Fig. 1 | A global virus–plankton interaction network shaped by latitude and temperature. a** Richness of viral communities at the stations of the *Tara* Oceans expedition (2009–2013). A total of 928 epipelagic metagenomes from 143 *Tara* Oceans stations are included in this study. Each pie chart represents the contributions to richness by six taxonomic main groups, and the size of the pie chart is proportional to the total richness at the station. The richness of two depths (Surface and Deep Chlorophyll Maximum) and different size fractions (Pico, Piconano, Nano, Micro, Macro, and Broad) are integrated into one pie chart. Dashed lines indicate the boundary of Polar samples. **b** A virus–plankton interaction network. Five individual networks inferred using input matrices for the relative frequencies of eukaryotes (five size fractions) and giant viruses (Pico-size fraction). The best positive or negative association (i.e., the edges with the highest absolute weights

between two genomes) were selected to build the integrated interactome. Node color represents the temperature optima of each genome for viruses and eukaryotes. A total of 1347 nodes (567 eukaryotes and 780 viruses) are in the network. Of these nodes, 1191 nodes (554 eukaryotes and 637 viruses) are colored according to their temperature optima. **c** The distribution of pairwise sequence similarity of proteins (one protein from the eukaryotic genome and one from the viral genome). Blue line indicates the distribution for pairs with a strong virus–eukaryote association in the interactome (edge weight of ≥0.4), while the red line is for pairs lacking a strong association. The two distributions are significantly different ($P = 1 \times 10^{-13}$, two-sided Wilcoxon signed-rank test). Source data are provided as a Source Data file.

GEVE-like signals (Supplementary Fig. 7c). Diatoms of Chaetocerotales are abundant and diverse in both the Arctic and Southern Oceans[51,52]. In the marine eukaryotic genome database[30], Chaetocerotales genomes exclusively distribute in high latitudinal polar oceans (temperature optima = 0.72 ± 0.63 °C). So, the putative

virus–Chaetocerotales relationship may account for the diversity of giant viruses in high-latitude regions. However, this in silico prediction is limited by the absence of direct evidence of host–virus interaction and by the current unavailability of genomes of polar Chaetocerotales isolates.

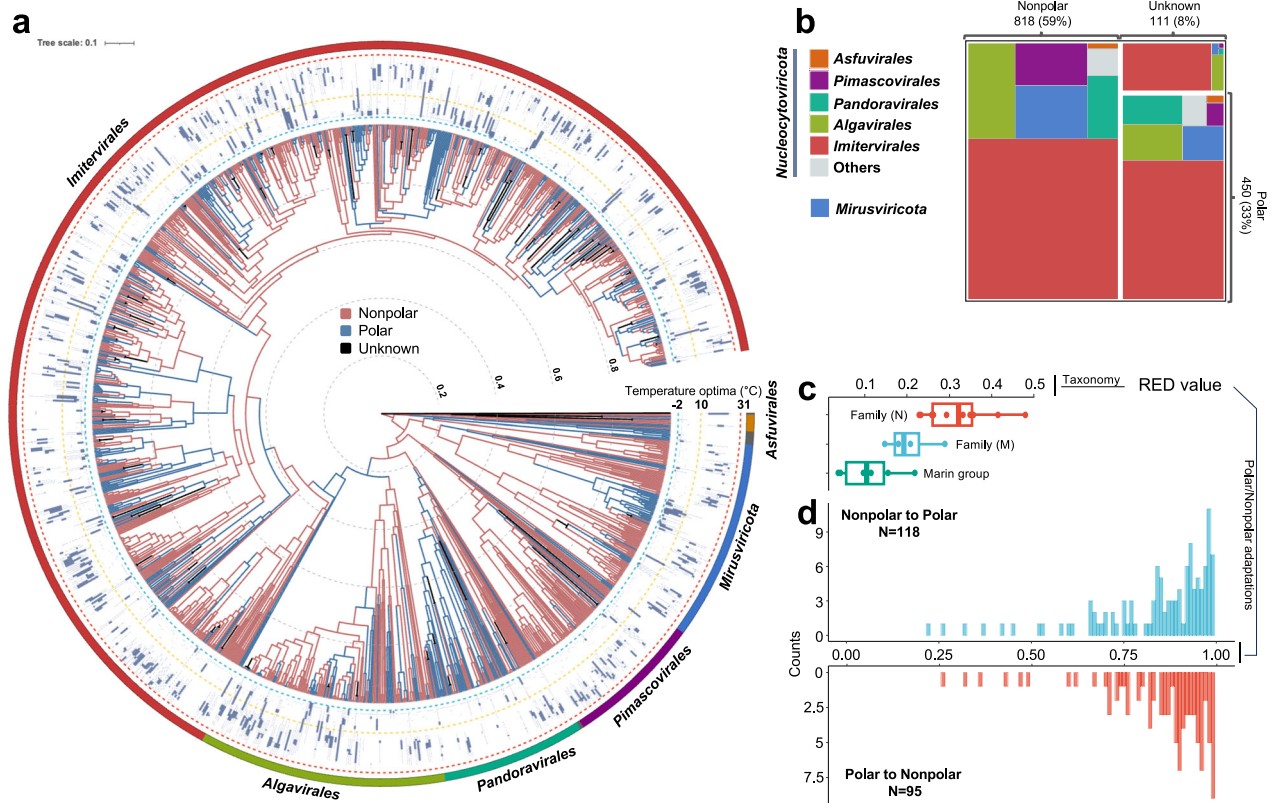

**Fig. 2 | Inferred ancestral polar and nonpolar niches for viruses. a** Ancestral "Polar" and "Nonpolar" states were estimated using the phylogenetic tree based on a one-parameter equal rates model. The outermost layer shows the taxonomy of six main groups. The boxplots in the second layer show the temperature optima of the viral genomes. For each box, $n = 10,000$ temperature values were analyzed as outlined in the methodology section on robust ecological optimum and tolerance. Only polar and nonpolar genomes were included in the tree. **b** The treemap diagram shows the number of viruses assigned to Polar, Nonpolar or "Unknown" biomes. Colors indicate the main taxonomic groups. **c** Relative Evolutionary

Divergence (RED) values for viral main groups ($n = 6$) and families. *N* stands for the phylum *Nucleocytoviricota* ($n = 17$) and M stands for *Mirusviricota* ($n = 5$).
**d** Histograms of RED values for the nodes at which "polar" or "nonpolar" adaptation events were inferred. RED values of child nodes in adaptation events were shown. For box plots, center lines show the medians; box bounds stand for the 25th and 75th percentiles; whiskers extend 1.5 times the interquartile range from the 25th and 75th percentiles; outliers are represented by dots. Source data are provided as a Source Data file.

## Recurrent polar adaptations throughout viral evolution

To investigate viral adaptation across the polar barrier, we assigned ecological niche categories, either "Polar" or "Nonpolar", to individual viral genomes. Of 1380 viral genomes, 450 genomes were classified as Polar, while 818 genomes were classified as Nonpolar (Fig. 2a,b). 111 genomes were labeled "Unknown" because of their ambiguous distribution patterns. This ecological niche assignment was consistent with the robust temperature and latitude optima (Supplementary Fig. 8a). All six genomes of *Proculviricetes*[31], a recently discovered class-level group recovered exclusively from the Arctic and Southern Oceans, were classified as Polar viruses as expected. A lineage of mirusviruses (clade M2[31]) formed a large clade mainly composed of Polar viruses with an additional sub-clade composed of Nonpolar viruses. Limitations, such as unequal sampling and sequencing depth, may potentially affect niche assignment. However, several cases justify our niches assignment using global-scale abundance profiles. For example, *Chrysochromulina ericina* virus, isolated from high latitude Norwegian coastal waters[53], was correctly assigned to the Polar niche. The Polar niche assignment to a metagenome-assembled genome (MAG) derived from Arctic samples was corroborated by its phylogenetic grouping with organic lake phycodnaviruses, which were independently derived from Antarctic organic lake metagenomes[15] (Supplementary Fig. 8b).

We then performed Polar/Nonpolar state reconstruction for ancestral nodes in the tree using a maximum likelihood approach (see Methods). As a result, 118 Nonpolar-to-Polar and 95 Polar-to-Nonpolar

niche adaptations were inferred along the branches of the tree (Fig. 2a). These adaptations thus occurred recurrently throughout the evolution of these viruses starting from the root of the tree, which was inferred as Nonpolar. Yet, our data could not exclude the possibility of a polar-origin scenario due to the difficulty in determining the root of the tree of giant viruses. The divergence of these viruses is estimated to predate the divergence of eukaryotes[22,24]. Most of the reconstructed niche adaptations occurred relatively recently after the formation of genera, but some adaptations were inferred to have occurred during the early stage of evolution, corresponding to order-level divergence (Fig. 2c,d).

## Polar-specific viral functions and their phylogenetic distributions

Genomic adaptation (i.e., adaptation by alteration of gene repertoire) to polar regions was investigated based on functions encoded in the viral genomes. We first annotated genes in the viral genomes with the KEGG Orthologs (KOs). For KOs ($n = 1591$) that were observed in more than four genomes, we calculated robust temperature and latitude optima (Supplementary Data 4). The temperature optima ranged from $-1.54\,°C$ to $27.31\,°C$, and the latitude optima from $5.25\,°$ to $78.96\,°$. The distribution of these values revealed two major groups of KOs: one distributed in high-latitude/low-temperature regions ($n = 314$, 19.74%) and another in lower-latitude/higher-temperature regions ($n = 1,277$, 80.26%) (Fig. 3a). The 314 Polar-specific genes had temperature optima below $10\,°C$ and latitude optima above $50\,°$. The temperature and

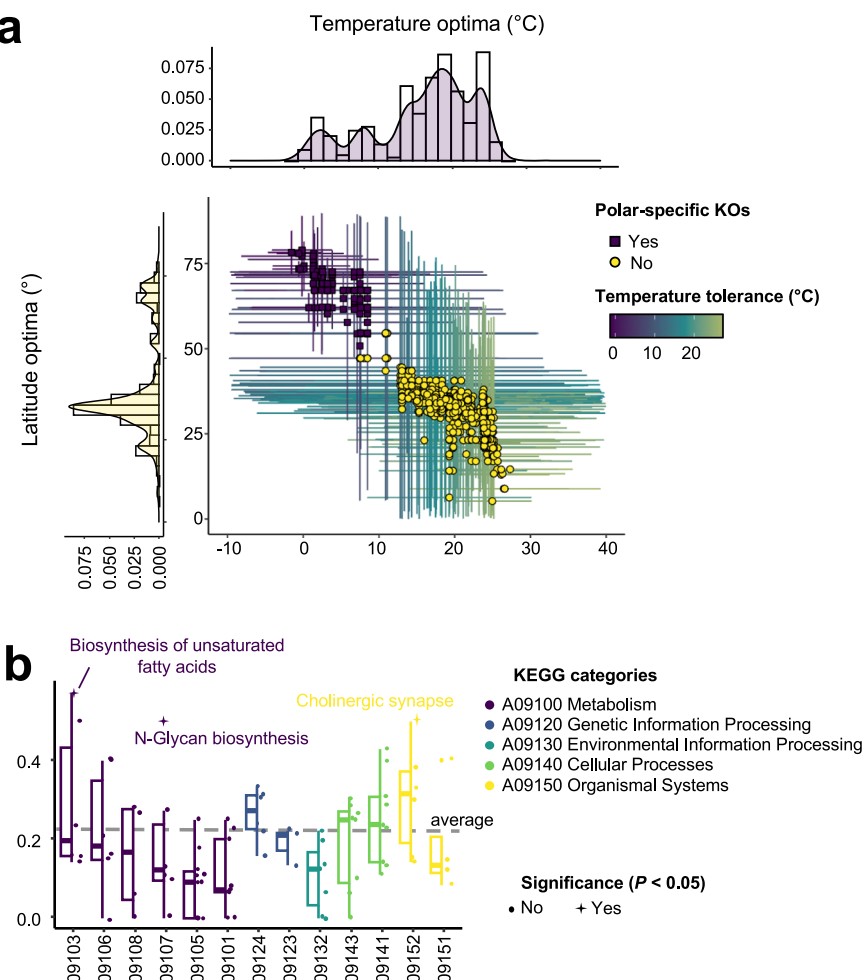

**Fig. 3 | Ecological niche of KEGG Orthologs (KOs) and polar-enriched pathways. a** Distribution of the temperature optima and latitude optima for KEGG Orthologs (KOs) found in viral genomes. Colors of dots represent the Polar or Nonpolar niche for each KO. Bars indicate the tolerance ranges of temperature (horizontal) and latitude (vertical). Histograms show the distributions of temperature and latitude optima. **b** A boxplot with jitter of ratio of Polar KOs in each pathway. Totally, *n* = 84 pathways were examined. Stars and labels correspond to pathways in which Polar KOs were significantly enriched (*P* < 0.05, one-sided Fisher's exact test) while circles stand for the non-significant pathways. *P* values of Biosynthesis of unsaturated fatty acids, N-Glycan biosynthesis, Cholinergic synapse are 0.03, 0.02, 0.04, respectively. Colors indicate the top categories of pathways in the KEGG database. For box plots, center lines show the medians; box bounds stand for the 25th and 75th percentiles; whiskers extend 1.5 times the interquartile range from the 25th and 75th percentiles. The overall ratio of Polar KOs to all KOs is indicated by a dotted line. The *x* axis shows the second-level categories: Lipid metabolism (09103); Metabolism of other amino acids (09106); Metabolism of cofactors and vitamins (09108); Glycan biosynthesis and metabolism (09107); Amino acid metabolism (09105); Carbohydrate metabolism (09101); Replication and repair (09124); Folding, sorting and degradation (09123); Signal transduction (09132); Cell growth and death (09143); Transport and catabolism (09141); Endocrine system (09152); Immune system (09151). Source data are provided as a Source Data file.

latitude optima for conserved core genes of giant viruses were found in the second group, being distributed at around 13–14 °C and 37–40 °C, respectively.

We then calculated the phylogenetic diversity of individual KOs using the viral phylogenomic tree as a reference to assess the breadth of their phylogenetic distribution (Supplementary Fig. 9a). Overall, Polar-specific KOs showed a relatively low phylogenetic diversity (median = 6.94) compared with other KOs (median = 9.67) (Wilcoxon rank-sum test, *P* < 0.01), indicating relatively narrow phylogenetic distributions of the Polar-specific KOs. To further characterize the phylogenetic distributions of the 314 Polar-specific KOs, we examined the strength of phylogenetic signals in their distribution using a model comparison approach (see Methods). This analysis revealed that the reference phylogenomic tree has insufficient explanatory power for the phylogenetic distribution of 193 Polar-specific KOs (61%) out of the 314 KOs (chi-squared test, *P* < 0.05). It is thus inferred that additional factors rather than speciation history impacted the phylogenetic distribution of these KOs;

environmental conditions or associated host distributions could be such factors.

## Polar-specific viral functions and metabolic pathways

The average proportion of polar-specific KOs among all genes with KO annotations in a viral genome was 15.84% for Polar genomes, which was significantly higher than Nonpolar (6.95%) and Unknown (7.93%) genomes (Supplementary Fig. 9b; Kruskal-Wallis test, *P* < 0.01). Among Polar-specific KOs, ceramide glucosyltransferase (K00720) and dihydrofolate reductase (K18589) were exclusively distributed in polar genomes. Ceramide glucosyltransferase catalyzes sphingolipid glycosylation, indicating the biosynthesis of viral sphingolipids may improve the fitness of polar viruses[54]. Dihydrofolate reductase could provide dTMP pools for low GC content viruses, and a possible role of this function is to facilitate the replication of viruses in the persistent infections[55]. Additionally, nitrate transporter (K02575) had a high ratio of polar to nonpolar phylogenetic diversity (ratio = 7.96), thus showing a comparatively wide phylogenetic distribution in Polar genomes. The

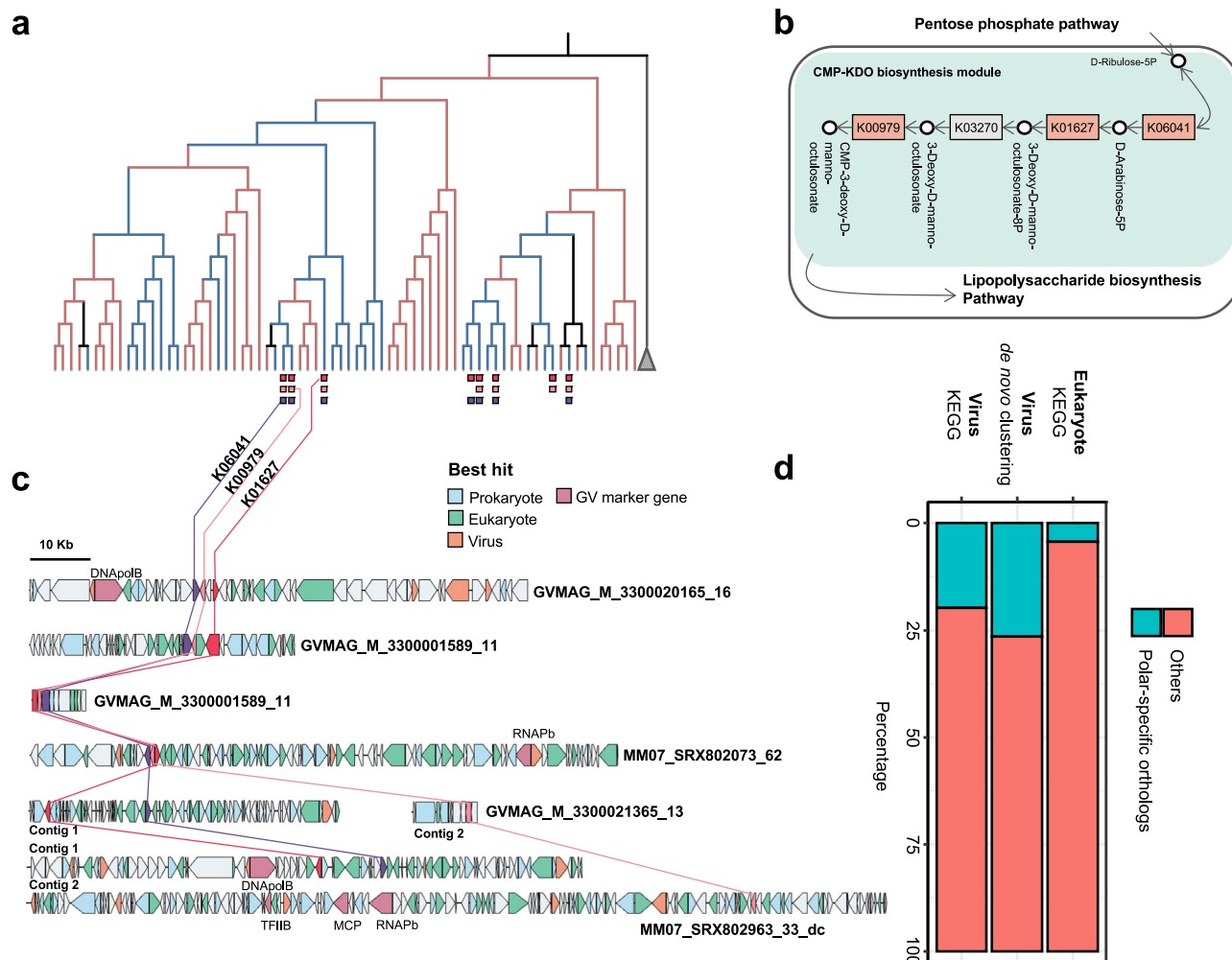

**Fig. 4 | Independent genomic adaptation of giant viruses.** 244 functions (KOs) were enriched at individual lineages. One example was given in (**a**), Three KOs that were present exclusively in more than five Polar genomes in a selected *Mesomimiviridae* clade. Three of them (K01627, K00979, K06041) were encoded in the same genomes and formed a near-complete CMP−KDO biosynthesis module shown in (**b**), Schematic of the three Polar enzymatic steps in the CMP−KDO biosynthesis module. **c** Genome maps of MAGs encoding three CMP-KDO KOs. Best matched taxonomies of genes are shown using the same colors, with the key provided at the top right. Colored lines connect detected CMP-KDO KOs between every two contigs. "contig1" and "contig2" indicate two contigs come from the same MAG. **d** Proportion of Polar and Nonpolar specific functions (KOs and GCCs) in viruses and eukaryotes.

nitrate transporter pathway has a role in assimilating extracellular nitrate/nitrite, implying a potential role for Polar viruses to reprogram host metabolism to fit the nitrate-deficient polar oceans[56]. Some metabolic functions, including CoA biosynthesis (4′-phospho-pantetheinyl transferase) and secondary metabolite biosynthesis (hydroxymandelonitrile lyase and 2-polyprenyl-6-hydroxyphenyl methylase), also showed a high phylogenetic diversity for Polar genomes.

At the pathway level, three pathways were significantly enriched with Polar-specific KOs (Fig. 3b; Fisher's exact test, $P < 0.05$). These were unsaturated fatty acid biosynthesis, N-glycan biosynthesis, and cholinergic synapse pathways. A high proportion of unsaturated fatty acids is known to be an adaptive trait among bacteria inhabiting low temperature and high pressure environments[57]. Eukaryotes and their viruses have similar membrane compositions to those of bacteria. Indeed, giant viruses isolated from high latitude areas encode enzymes for the biosynthesis of unsaturated fatty acids[44], which may be part of a strategy to rewire the host fatty acid physiology[54]. N-glycan plays an important role in the virus replication cycle, including virus recognition and virus release, and potentially contributes to the stability of virions[58]. The enrichment of Polar-specific KO in cholinergic synapse (albeit its pathway name reflects biology of animals and may not be

relevant to unicellular eukaryotes) implies the ability of polar viruses to regulate signal transduction.

**Other potential polar adapted functions**

In addition to the above statistical analyses based on the temperature and latitude optima, we performed an enrichment analysis of KOs by examining their presence in Polar and Nonpolar genomes at different evolutionary scales to capture a variety of situations in the phylogenetic distributions of KOs. Specifically, this analysis was performed at four different lineage levels (i.e., root, main group, family, and genus). The analysis revealed 265 functions that were significantly enriched in Polar genomes inside at least one lineage (Fisher's exact test, $P < 0.05$; Supplementary Data 4). As expected, KOs enriched in Polar viral genomes showed lower temperature optima than other KOs (Supplementary Fig. 9c; Wilcoxon rank-sum test, $P < 0.01$). For a finer-grained observation, we focused on one *Mesomimiviridae* clade, containing a similar number of Polar ($n = 32$) and Nonpolar ($n = 40$) genomes scattered in a subtree of the phylogenomic tree. In this example, four functions were found in more than five genomes from different Polar clades (Fig. 4a). Three of them (K01627, K00979, K06041) co-occurred in the same genomes and formed a near-complete CMP−KDO biosynthesis module in the lipopolysaccharide biosynthesis pathway

(Fig. 4b). A large proportion of other genes encoded in the contigs harboring the three CMP−KDO KOs best matched to viral genes (averaging 11% and up to 45%; Fig. 4c). Furthermore, four contigs encoded NCLDV marker genes. These data confirms that the identified CMP-KDO KOs are bona fide viral genes. Enzymes of CMP−KDO biosynthesis were previously found encoded by a giant virus and suggested to add glycoconjugates to the surface of virions to enhance virion-cell recognition[59]. The genomes in the examined Polar clade may also coat virions with glycoconjugates, to potentially enhance virus−host interactions and/or virion stability.

The KO system can annotate only functionally known genes, and therefore we calculated robust temperature and latitude optima for gene cluster communities, de novo clusters of viral genes[31]. The result showed a slightly higher proportion of Polar-specific gene clusters (26.43%) than obtained by KO annotations (19.74%) (Fig. 4c; Supplementary Fig. 10a), indicating the presence of genes of unknown function that have Polar-specific distributions. We also found that Polar genomes have a slightly but significantly higher proportion of Alanine-rich low-complexity regions than Nonpolar and Unknown genomes (Supplementary Fig. 9d; Dunn's test, $P < 0.05$, following a significant Kruskal-Wallis test, $P = 0.0002$). These low-complexity sequences potentially have an anti-freeze function, as alanine-rich helical structure is one of the significant characteristics of type I antifreeze proteins for ice growth inhibition[60]. Additionally, the proportion of Polar viral genomes that encoded antifreeze protein homologs ($n = 7$, 1.6%) was higher than the genomes of other groups ($n = 6$, 0.65%), although the difference was not statistically significant ($P > 0.05$).

### Polar-specific functions in microbial-eukaryotes
Finally, to examine whether genomic adaptation occurs in eukaryotic plankton in polar regions and to test if the adaptation is related to the one in viruses, we calculated the temperature and latitude optima for KOs ($n = 11,988$) assigned to genes in eukaryotic genomes. A similar pattern of Polar and Nonpolar KO groups was identified, although the proportion of the Polar KO group ($n = 523$, 4.36%) was much smaller than that for viruses (19.74%) (Fig. 4c; Supplementary Fig. 10b). Interestingly, of the 523 KOs in the eukaryotic Polar group, only four were found in the viral Polar group. These were PPM family protein phosphatase, L-galactose dehydrogenase, transcription factor S, and ATP-dependent DNA helicase DinG. This result indicates that most Polar viral functions do not exhibit the same temperature/latitude optima as their homologs in eukaryotic genomes, suggesting that genomic adaptations are uncoupled between viruses and eukaryotes.

For a virus to adapt to a new environment, it is a prerequisite that its host already adapts to the environment. This host adaptation would give rise to additional environmental (or micro-environmental) changes for the virus. Such micro-environmental changes include alterations of cell surface structures as well as intracellular metabolic states. Virus-host interactions involve different processes such as adhesion to the cell, metabolic remodeling, viral genome replication, genome packaging and egress from the cell. These processes are likely affected not only by ambient physico-chemical conditions (such as temperature) but also, and more profoundly, by the biochemical and physiological conditions of the host cell that adapts to the target environment. Therefore, for a virus to adapt to a new environment, it needs to cope with both environmental changes and environment-induced host cell alterations. Our results suggest that large and giant DNA virus adaptation to polar environments requires alteration or innovation of viral metabolic strategies, which is manifested in viral genomic changes. In this adaptive process, viruses appear to take their own strategies that are distinct from the host strategies for their adaption in the same habitat.

The adaptation of cellular organisms to their environments could be largely manifested in their functional repertoire. Previous

discoveries of presence of ecologically significant genes (such as lipid metabolism and rhodopsin) in viruses[54,61] indicated that functional repertoire could also be important for adaptive evolution of viruses. However, the functional adaptation of viruses at a wide geographic scale has not been investigated as deeply as for cellular organisms. Thanks to the recent progress in metagenomics, we investigated the links between the biogeography, host types, and gene repertoire of large and giant DNA viruses infecting marine eukaryotes. We confirmed the existence of a strong polar/nonpolar barrier for these viruses and revealed size fraction-dependent Arctic diversity hotspots for some virus groups, which may reflect a high diversity of their hosts in cold environments. Temperature was an important factor that shaped the virus−host interactions of polar environments. Consistent with these findings, our analyses implied some potential polar host, such as diatoms, could contribute to the polar distribution of giant viruses. Our phylogenomic tree and ancestral state reconstruction revealed back-and-forth adaptations between lower- and higher-temperature niches that occurred recurrently throughout the long evolutionary course of these viruses. Numerous functions, especially ones related to host interactions, were found to be specific to viral polar adaptation, but most of them were not identified as polar-specific functions in eukaryotes, suggesting a decupling of viral and host polar adaptations. Furthermore, the gene repertoire of these large DNA viral genomes appears more evolutionarily flexible and responsive to temperature change than that of eukaryotic genomes.

## Methods
### Global Ocean Eukaryotic Viral (GOEV) database
Metagenomic datasets and environmental data are provided in Supplementary Data 1. The Global Ocean Eukaryotic Viral (GOEV) database contained 1817 viral genomes[31]. The initial version of the GOEV database included 697 genomes reconstructed from 798 *Tara* Oceans metagenomes, 1187 MAGs from two previous metagenomic surveys[33,62], and 235 reference NCLDV genomes. Redundancy in the dataset was reduced based on Average Nucleotide Identity (ANI) of 98% (by always retaining the 697 genomes from *Tara* Oceans metagenomes), ultimately resulting in a refined database containing 1817 genomes. Taxonomic inference, read mapping, gene call and gene annotation of the GOEV were performed in a previous work[31]. 1380 detected viruses were classified into six main taxonomic groups: five orders (i.e., *Algavirales*, *Asfuvirales*, *Imitervirales*, *Pandoravirales*, and *Pimascovirales*) and the newly discovered phylum, *Mirusviricota*. Metagenomes of six different size fractions were used in this study: 0.22–1.6 μm or 0.22–3.0 μm ("Pico"), 0.8–5 μm ("Piconano"), 5–20 μm ("Nano"), 20–200 μm ("Micro"), 200–2000 μm ("Macro"), and 0.8–2000 μm ("Broad"). The size fraction below 0.22 μm was excluded in this study because of the low relative abundance and high overlap with species from the Pico size fraction. The mapping was carried out with BWA v0.7.15 (minimum identity of 90%). MAGs results were retained if at least 25% of the viral genome was covered by reads. Relative abundance of a giant virus in each sample was calculated in Reads Per Kilobase per Million mapped reads (RPKM). We did not perform subsampling prior to biodiversity calculation because sequencing depths of metagenomes did not significantly influence the Shannon's index of giant virus communities or to the extent latitude did.

### Phylogenetic tree construction
Phylogenetic trees used in this study were reconstructed using IQ-TREE v.1.6.2[63]. The viral species tree was reconstructed with the site-specific frequency PMSF model following a best-fitting model according to the BIC from the ModelFinder Plus option. Tree visualization and analysis were carried out using ETE3 toolkit v.3.1.1[64]. iTOL v.6 was used to visualize the phylogenetic trees[65]. Phylogenetic

diversity was calculated using the 'pd' function in the R package 'picante'[66] for polar and nonpolar genome subsets.

## Ecological analyses

Diversity analyses were performed using R v.4.0.1[67] in Rstudio v.1.3.959[68]. To evaluate the diversity of each sample, the richness (number of MAGs), Shannon's index and Pielou's evenness were calculated with the package "vegan"[69]. Compositional variation among samples was assessed with a non-metric multidimensional scaling (NMDS) ordination based on Bray−Curtis dissimilarity. Samples with low viral abundance and richness produce outliers that reduce the readability of the NMDS ordination plot. To avoid such a bias, samples for which the sum of cumulative coverage was less than 10 or richness was less than 5 (set as the cutoff threshold) were removed from the compositional variation analyses. Statistical significance of differences among the sample groups (size fractions and biomes) was tested using an ANOSIM (analysis of similarities) with 9999 permutations. The significance threshold was set to a $p$ value of 0.01. The plots and maps of sampling stations were generated by packages "ggplot2"[70] and "rgdal"[71].

## Gene annotation and clustering

Genes were predicted using Prodigal v.2.6.3[72] within anvi'o v6.1[73] with the default parameters. Gene cluster communities were classified through the AGNOSTOS[74] workflow. Those two steps were performed and described in a previous work[28]. For functional annotation, genes were assigned to KEGG Orthologs (KOs) using eggNOG-mapper v.2.1.5[75] ("Diamond" with an E-value cut-off of $1.0 \times 10^{-5}$). Viral marker genes were searched with in-house HMM profiles from NCVOG (nucleocytoplasmic virus orthologous genes)[76] and GVOG (giant virus orthologous groups)[23] databases using HMMER v.3.2.1 (http://hmmer.org) with an $E$ value of $1 \times 10^{-3}$. Antifreeze proteins were detected using InterProScan v.5.44-79.0[77]. Low-complexity regions of protein sequences were identified using the option '-qmask seg' in usearch v.11.0.667[78].

## Virus−plankton interaction network

We determined the relative abundance matrix for the virus MAGs from the Pico size fractions and relative abundance matrices for eukaryotic MAGs from five cellular size fractions (Piconano, Nano, Micro, Macro, and Broad). To create the input files for network inference, we combined the viral matrix with each of the eukaryotic matrices (corresponding to different size fractions), and only the samples represented by both viral and eukaryotic MAGs were placed in new files. Relative abundances in the newly generated matrices were normalized using centered log-ratio (clr) transformation after adding a pseudo-count of one to all matrix elements because zero cannot be transformed in clr. Normalization and filtering were separately applied to viral and eukaryotic MAGs. We then removed the MAGs that had fewer than three sample observations. Network inference was performed using FlashWeave v.0.15.0[79] with Sensitive mode to set a threshold of $\alpha < 0.01$ as the statistical significance and without the default normalization step. All detected pairwise associations were then assigned a weight that ranged between −1 and +1. The network was visualized with Cytoscape v.3.7.1[80] using the prefuse force-directed layout. Proteins between linked genome pairs were aligned using BlastP in Diamond v.2.0.6[81] with an $E$ value cut-off of $1.0 \times 10^{-50}$.

## Host prediction

First, we pooled network associations from five size fractions by keeping the best positive or negative associations (i.e., the edges with the highest absolute weights). We used a phylogeny-guided filtering approach, TIM[40], to predict the host using the global nucleocytoplasmic large DNA virus (NCLDV)−eukaryote network. TIM assumes that evolutionarily related viruses tend to infect evolutionarily related hosts[82]. All the virus−eukaryote associations were mapped on the viral phylogenetic tree to calculate the significance of the enrichment of specific associations using TIM. If a specific eukaryotic group (NCBI taxonomies, including order, class, and phylum) was significantly enriched under a node of the viral tree, compared to the rest of the tree (determined using Fisher's exact test and Benjamini−Hochberg adjustment), that specific eukaryotic group was deemed the predicted host. The result was visualized with iTOL v.6.

## Endogenous viral signals

We searched the viral signals in 713 genomes from the eukaryotic environmental genomes database using VirSorter2 v.2.2.3[30,83]. Both NCLDV and *Lavidaviridae* (virophage) genomic insertions (or co-binning) were searched using --min-score 0.85 and 0.95 for NCLDV and virophage, respectively. We next obtained long-read assembled genomes of two *Chaetoceros* isolates, *C. muelleri*[84] and *C. tenuissimus*[85]. GEVEs were detected using ViralRecall v.2.1 (-s 0 -w 15)[86].

## Size index

Each *Tara* Oceans metagenome corresponds to a specific filtering size fraction (Pico, Piconano, Nano, Micro, Macro, and Broad size fractions), which were sorted as a list by increasing size. The size index, corresponding to the filtration fraction, was designed in this study and serves as an indicator of the host range, with a larger size index implying that the virus infects larger-bodied organisms. An index constant was set for each size fraction from small to large: Pico = 1, Piconano & Broad = 2, Nano = 3, Micro = 4, Macro = 5 (the Broad and Piconano size fractions were merged because of their similar relative abundances and lack of Arctic samples for the Piconano fraction). We calculated the size index for a given genome by first multiplying the RPKM of the genome in a sample by the corresponding index constant, then dividing the sum of the products by the overall sum of the RPKMs of the genomes from all samples.

## Biome and size niche

Each sample was associated with one specific marine biome (Coastal, Trades, Westerlies, or Polar). To straightforwardly investigate the difference between polar and nonpolar regions, we labeled Coastal, Trades, and Westerlies samples as "Nonpolar". First, we assigned genomes with zero mapping signals in "Nonpolar" metagenomes to "Polar" biome niche. Likewise, we assigned genomes with zero mapping signals in "Polar" metagenomes to "Nonpolar" biome niche. Additionally, utilizing RPKM profiles, we determined the statistical significance to assess whether a specific genome is overrepresented in either "Polar" or "Nonpolar" metagenomes using the Wilcoxon rank-sum test. The Benjamini−Hochberg (BH) correction was applied to significance values to account for the effect of multiple testing[87]. The significance threshold was set to a corrected $P$ value of 0.05. Similar assignments were performed for two size fractions: intercellular (Pico-size) and intracellular (Piconano, Nano, Micro, Macro, and Broad).

## Robust ecological optimum and tolerance

We calculated the robust ecological optimum for a genome (or a gene), which reflects the optimal living condition regarding a given environmental parameter and a tolerance range around this optimum defined by lower and upper bounds[35,36]. This metric has been applied to microeukaryote plankton[35]. Given the polar barrier for giant viruses, which was described previously[19] and confirmed in this study, we considered this metric is also useful to characterize ecological properties of viruses studied here. For each genome (or a gene), we computed the proportion of RPKM in a given sample relative to the sum of RPKM over all samples. We then used these proportions to populate a weighted vector of a fixed size ($n = 10,000$) with environmental values accordingly. For example, if the proportion of RPKM for Genome1 in sample1 represents 5% of the Genome1's cumulated RPKM across all

samples, then 5% of the elements of the weighted vector will be filled with the environmental value measured for sample1 (e.g., temperature and latitude of sample1).

The ecological optimum is then defined as the median value (Q2) of this vector, and the tolerance (niche) range is given by the interquartile range (Q3 to Q1; some environmental parameter values were missing [nonavailable (NA)] for some samples). To avoid inferring spurious ecological optima and tolerance ranges for genomes (or genes) for which there were many missing values, we set a cut-off of 10 observations with non-NAs and a minimum fraction of 30% non-NA values[35].

### Ancestral states estimation and relative evolution divergency
Ancestral states of Nonpolar and Polar viruses were estimated using the function "ace" (Ancestral Character Estimation) in the R package 'ape'[88]. The input files were a rooted phylogenetic tree based on the four-hallmark gene set (RNApolA, RNApolB, DNApolB and TFIIS). In the tree, we retained only viruses with biome assignments of Polar or Nonpolar, and excluded viruses with "Unknown" biomes. We used type = "discrete", method = "ML", and model = "ER" (one-parameter equal rates model). The ancestral states were analyzed based on a series of likelihood values for Polar and Nonpolar. Relative Evolutionary Divergence values were calculated using the "get_reds" function in the package "castor"[89].

### KO enrichment in Polar viral genomes and pathways
"Polar", "Nonpolar", or "Unknown" biome niche was assigned to each viral genome based on presence/absence and overrepresentation ("*Biome and size niche*" section). For individual lineages at four taxonomic levels (root, main group, family, and genus), the enrichment of a given KO in Polar genomes assessed using Fisher's exact test in SciPy v.1.7.1[90]. Adjustments for multiple testing were performed using the Benjamini-Hochberg (BH). The significance threshold was set to a corrected $P$ value of 0.05.

Polar-specific KOs were defined as those with a temperature optimum below 10 °C and a latitude optimum above 50°. For pathways with at least half of the detected KOs as polar-specific KOs, we compared the fraction of components (i.e., enzymes) defined as polar-specific KOs with the fraction of all other pathways. This fraction was tested by the Fisher's exact test and adjusted by the Benjamini-Hochberg (BH). This analysis excluded rare KOs (observed in fewer than five genomes). To avoid the enrichment of pathways led by sparse KOs, the pathways were exhibited only if the number of detected viral KOs in a given pathway constituted more than 10% of the total number of KOs in the pathway.

### Phylogenetic signal of functions
We hypothesized that the phylogenetic distributions of some polar specific functions (i.e., "trait distribution") could be better explained in part by environment selection rather than only by speciation history. We therefore compared two models, (i) the Brownian motion model (Pagel's lambda = 1, used as the null hypothesis in which the distribution of a trait is simply explained by species tree) and (ii) the Lambda model (0 ≤ Pagel's lambda ≤ 1; lambda = 0 corresponds to the lack of phylogenetic signal in the distribution of a trait), by the likelihood ratio test using the function "fitContinuous" in an R package "geiger"[91]. The $p$ values to reject the null hypothesis were calculated by assuming chi-squared distribution with 1 d.f. for the likelihood-ratio test statistic and adjusted using the BH procedure. The threshold was set to a corrected $p$ value of 0.05

### Reporting summary
Further information on research design is available in the Nature Portfolio Reporting Summary linked to this article.

## Data availability
The metagenome data from *Tara* Oceans are available at the ENA under accession PRJEB402, the metadata of metagenomes used in this study are summarized in Supplementary Data 1. FASTA files for the 1817 giant virus genomes from the Global Ocean Eukaryotic Viral (GOEV) database can be accessed here: https://doi.org/10.6084/m9.figshare.20284713. Additionally, the accession numbers of 1593 non-redundant marine *Nucleocytoviricota* and mirusvirus MAGs and 224 reference genomes in the GOEV database are provided in Supplementary Data 2. Other data used in this study include: Giant Virus Orthologous Groups (GVOGs) database (https://faylward.github.io/GVDB/); Virus-Host Database (https://www.genome.jp/virushostdb); *Tara* Oceans Eukaryotic Genomes Database (https://www.genoscope.cns.fr/tara); NCBI database (https://www.ncbi.nlm.nih.gov/genome). The data utilized in this study can be accessed from GenomeNet at: https://www.genome.jp/ftp/db/community/tara/PolarAdaptaiton/data/. Source data are provided with this paper.

## Code availability
The script used to calculate robust ecological optima is available at https://github.com/LingjieEcoEvo/PolarAdaptaiton/tree/main/optimum

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

## Acknowledgements

This work was supported by JSPS/KAKENHI (18H02279 and 22H00384, to H.O.), and the Collaborative International Joint Research Program of the Institute for Chemical Research, Kyoto University (No. 2021-29, 2022-26 to T.O.D.; No. 2022-27, to S.C.), and the H2020 European Commission project AtlantECO (award number 862923, to S.C.). Computational time was provided by the SuperComputer System, Institute for Chemical Research, Kyoto University. We further thank the *Tara* Oceans consortium, and the people and sponsors who supported *Tara* Oceans. *Tara* Oceans (including both the *Tara* Oceans and *Tara* Oceans Polar Circle expeditions) would not exist without the leadership of the *Tara* Expeditions Foundation and the continuous support of 23 institutes (https://oceans.taraexpeditions.org). This article is contribution number 147 of *Tara* Oceans. We thank Gabe Yedid, PhD, from Edanz (http://jp.edanz.com) for editing a draft of this manuscript.

## Author contributions

L.M. and H.O. designed the study. L.M. performed the primary biogeographical analysis. T.O.D completed the genome-resolved metagenomic analysis. M.G. performed phylogenomic analyses. E.P. generated the reads mapping data. A.F.-G. provided de novo clusters of viral genes. S.C., R.Y.N., J.W., H.K., H.E. contributed to the bioinformatics analysis. All the authors contributed to interpreting the data and writing the manuscript.

## Competing interests

The authors declare no competing interests.
