## [Peer Review File · Nature Communications]

Genomic adaptation of giant viruses in polar oceansREVIEWER COMMENTS

Reviewer #1 (Remarks to the Author):

This research article provided an insightful analysis of the divide between polar and nonpolar giant viruses (NCLDVs) in the Tara Oceans dataset. Through a variety of bioinformatic approaches, it was shown clearly that this division exists and there are several adaptations that polar viruses may have to inhabit this area. The enrichment analysis of KO groups, something commonly done in human RNAseq analysis, was a good approach to show differences and a highlight of the article. The classification of ecological optimum and polar vs non-polar viruses through taxonomy and abundance was also a novel approach that deserves praise. There do seem to be some issues with assumptions made when comparing RPKM across samples on top of some minor issues that need to be taken care of. Overall, this study pushes forward our knowledge of understudied polar viruses and sets forward a methodology that could be used for further classifications and analyses.

My comments and suggestions are provided below:

General introduction: The authors switch between talking about virus adaptations and virus-host adaptations which seem like two different things. The authors should focus on one of these two categories or talk about them in separate paragraphs.

Ln 39-44: This introduction seems overly broad and certain word choices are questionable such as "lush" and "nourish". I also think showing polar bear adaptations is a bit irrelevant in a paper about viruses.

Ln: 54: The sentence "A large proportion of Arctic-specific genes from these viruses were suggested under positive selection based on their mutation patterns." needs to be reworked as I am not sure what it is saying. The next sentence also could use rework and a citation of where the information is coming from.

Ln 54-62: The wording used seems to imply that it is the viruses themselves adapting to the environmental stressors when most of the papers you cite refer to the virus-host system as the unit doing the adapting. It might be useful to elaborate a bit on how viruses themselves are adapting - as the adaptation of viruses is likely a direct consequence of the adaptation of their hosts?

Ln 76: Mirusviruses are brought up without any introduction. It would be helpful to introduce them and why you are focusing on them specifically.

Ln 87-92: A bit more detail on the GOEV will be helpful here, although the authors included the citation. For example, authors should include how many of the genomes in these dataset were assembled from Tara Oceans dataset. If most genomes in GOEV were assembled from Tara Oceans data in the first place, it is expected that these genomes will have coverage from at least one Tara Oceans sequence library.

Ln 97: Do you mean R^2 ? If not, the author may want to explain what R-value is.

Ln 165: The author mentions that the relationship is unidentified but they just identified it. Also, given the biology of most diatoms (the silica shell), and lack of any direct evidence of host virus interaction in culture or genomic signatures, I would be cautious about inferring diatom-NCLDV host associations. The authors might want to add a caveat here.

Ln 178-180: this argument seems circular as viruses were assigned to categories based I'm assuming in part on isolation location. Validating this metric would require new viruses that weren't part of your training dataset. It seems self-evident that a virus only isolated in Arctic oceans would be classified as polar. I might be wrong in my interpretation, so please provide further discussion to address this concern.

Figure 3: The black circles around the circles are hard to see. I would make this bolder or come up

with a different symbol to visualize it.

Ln240: Again this seems circular to say that polar exclusive KOs are more abundant in Polar genomes if those genomes are where you most likely got the proteins from in the first place.

Ln 256-267: This paragraph could benefit from more flow between sentences as sentence structure is quite repetitive.

Ln 264: The most enriched should be mentioned first.

Ln 276: Seems self-explanatory like the above comments on Polar genomes showing lower temperature minima.

Ln 283-287: Mentioning bacteria here is odd as these viruses infect eukaryotes. This seems like a stretch to claim they are surviving by coating their capsid with these bacteria-like proteins without in-situ data. This is a strong claim to make, and I would be cautious about such a general assumption.

Ln 326: This is not a good hook sentence for the conclusory remarks. Functional repertoire doesn't feel like a "trait".

Ln 336: I would change presently to "previously" as you characterized it in this study.

Ln 345: It seems like a stretch to make claims about climate change here as it wasn't really the focus of any of your discussion or introductory material.

Ln 359: I might be wrong, but as far as I understand, mean coverage wasn't what was converted, raw read counts were.

Ln 360-362: It would seem that comparing RPKM across all 900+ samples is a rather 'bold' approach as RPKM can vary greatly if total DNA is different as well as sample distribution (<https://www.ncbi.nlm.nih.gov/pmc/articles/PMC7373998/#:~:text=RPKM and TPM represent relative,are close to each other.>).

While I understand that this normalization was perhaps necessary, it can result in misleading conclusions in my opinion. RPKM works best when the result is compared within the sample (i.e., which virus is most abundant within a sample compared to others?) - but comparing RPKM across such a large number of samples across this vast ecosystem can be tricky. The authors should discuss potential caveats of using RPKM and how that could potentially bias the interpretation of the results. It is also possible that I am missing something obvious, but further discussion should clarify these issues. For example, maybe RPKM is fine for 'correlation' purposes?

Ln 369: What does "tree structure manipulation and analysis" mean? The author should explain what was done here for the sake of reproducible research.

Ln 446-450: This section should be explained further as it is confusing why this was done. Previous methodology should be cited as to why it was calculated this way.

Ln 456-457: This part needs further explanation in terms of the methods.

Ln 467: See the above comment about RPKM but I do not think it is valid to compare it across samples without any normalization. This section could also be explained better as it seems critical to the thesis of the paper. I'm not sure if ecological optimum is biologically relevant to viruses and if so, a paper should be cited to back up why you are calculating it. Otherwise, the authors should further discuss the relevance of this metric in the context of their research and discuss if it is a novel concept in regards to virus ecology.

The author brings up Mirusviruses in the abstract and briefly in the introduction but then doesn't talk about them at all in the analysis. This group should either be talked about or removed from the study.

Reviewer #2 (Remarks to the Author):

Meng et al present a metagenomic analysis of giant viruses in the ocean, with particular emphasis on comparing polar viruses to those found in lower latitudes. They perform a suite of computational analyses to demonstrate that giant virus communities are distinct at higher latitudes and that polar viruses encode unique genes and have undergone the transition to polar climates multiple times independently. Overall this is a comprehensive study that examines giant virus communities in polar marine communities that will interest a broad range of readers. The figures are compelling and the bioinformatic analyses are well done. In general I would recommend adding more details for some of the methods, and I have some specific questions regarding some of the informatics and a few other suggestions to potentially strengthen the manuscript.

Perhaps I missed it somehow but it is unclear to me how the niche assignments were made. On Line 487 it is stated that this was done as previously described, but no citation is given. More detail on this would be welcome because it is an important step that underlies many subsequent analyses.

Line 29 - Can we be sure that temperature is driving these patterns, and not other features that are correlated with temperature? I.e. nutrient levels may be higher in polar oceans in the summer compared to lower latitude samples. In general this may be a point that is worth clarifying in other parts of the manuscript. In my experience many variables in marine datasets co-vary, so identifying the true drivers can be difficult.

Line 66 - "active" - please consider citing <https://doi.org/10.1128/mSystems.00293-21>. I believe this is a clear demonstration of high giant virus activity in marine systems.

Line 147 - it would be useful to know the percent of viral clades that have a predicted host according to this method. Right now it is hard to gauge how successful this approach was at predicting hosts. Also wouldn't host prediction occur at the genome level, and thus should be reported as such? E.g. XX percent of total MAGs had a predicted host according to this method. In general the paragraph starting on line 145 is a bit short, and I feel that more details could be added to improve readability.

The endogenous viral signals found in diatom genomes are quite interesting. I believe the writing of this section could be clarified so it is easier to understand whether the discussion is focusing on the endogenous PolBs recovered from the two previous studies cited, or whether this conclusion is primarily from the insertion signals identified here from the Delmont et al data. Moreover, the tree in ED Fig 7c is not entirely clear to me - the asfuvirales appear to cluster within the Imitervirales here, and both chaetoceros sequences are clustered together inside of that, so I'm not sure how a clear order-level designation can be made.

Regarding the insertion signals found in the Delmont et al data, it would be useful to clarify what exactly these are in the main text, even though additional details are in the methods. Based on the methods it seems that mostly metagenome-derived eukaryotic genomes from the Delmont et al study were used for this analysis, in which case it would be useful to confirm that the viral polBs are encoded on eukaryotic contigs (i.e. truly endogenized) and not viral contigs that may have been co-binned. Is it possible the viral PolB genes originate from viral contigs that were co-binned with the eukaryotic genomes? I understand that some details for this may be in the other paper, but given the importance of these claims it is worth providing details here. ViralRecall allows for contig-level plots to be generated (-f command), and so some contig-level VR plots showing the overall scores would be very useful here just to confirm they are chimeric (i.e. partly eukaryotic host, partly endogenized virus).

Figure 2- has this tree been rooted at the Asfuvirales? It is somewhat difficult to see so I would just like to confirm. The RED index may yield misleading results depending on the root placement.

Line 377 - when calculating alpha diversity, were the reads from the metagenomes sub-sampled? The different TARA stations have different sampling depths and this drastically impacts the alpha diversity estimates, so read rarefaction is needed for comparative purposes.

For the functional enrichment analysis, it would be good to see a supplementary spreadsheet with each KO and KEGG together with the counts used for statistics and the p-values. I couldn't find this- apologies if I missed it.

Figure 3b - I am not sure what the size of the bubbles indicates - the scale is shown but the units are absent.

Regarding the discussion on lines 256-267, is it possible that some pathways are significant due to few or individual KO enrichment, rather than presence of the entire pathway? I have noticed that with pathway-specific enrichment I often get false-positives because a small number of pathway components are present in high abundance, even though the others are missing. For example, a divergent acyl coa reductase may be highly abundant and make all of fatty acid metabolism look enriched, which would be misleading. For this reason I usually perform KO-specific enrichment and then look to see if the enriched KOs tend to belong to the same pathway. For example, are other genes in sphingolipid metabolism enriched in polar MAGs, or just the sphingolipid glycosyltransferase? Sphingolipids are likely involved in apoptosis stimulation as a means of viral egress (probably worth citing Vardi et al Science here [10.1126/science.1177322](https://doi.org/10.1126/science.1177322)), so it would be interesting to know if this was a common strategy for egress in polar viruses.

I feel like the presence of the "neuroactive ligand receptor interaction" is a bit strange and possibly a false-positive, perhaps because some ankyrin repeat proteins have spurious matches to proteins in the KEGG database. Manual annotation of some of the proteins that are driving this pattern could reveal more detail and confirm this observation.

Line 287 - "to enhance their interactions with Polar hosts." - is it possible it also plays a role in virion stability?

Would it be possible to make a table of the most enriched KOs in polar viruses? It would be useful to present a table of the top 10 or 20 most enriched genes. Given the emphasis on viral genomic adaptations I feel that the KEGG analysis may deserve more weight.

It was difficult for me to interpret the supplementary spreadsheets because the column headers were not described. I usually include a tab in the spreadsheet that includes descriptions of the column headers, and that may be useful here. For Supplementary Table 4 I could not see p-values for the statistics, and I was hoping to find the 314 polar-specific genes mentioned in the text (line 212).

I don't quite understand the difference between ED Fig 1d and ED fig 5. Some additional explanation in the legends would be welcome. In ED 5c it appears the diversity of Imitervirales increases markedly at higher latitudes, but this is not the case in ED Fig 1d. Is there a mid-latitude peak in giant virus diversity? It would be very interesting if there was a bimodal diversity peak.

Review written by Frank Aylward

Reviewer #3 (Remarks to the Author):

Meng et al. analyzed the gene repertoires of giant virus genomes, including metagenome-assembled genomes (MAGs) from global oceans, and inferred the existence of an ecological barrier between polar and nonpolar giant virus communities, with the former having genes/functions that indicate their „genomic adaptation“ („adaptation by alteration of gene repertoire“) to polar

environments in a way that is independent of their host adaptation. The adaptation of marine giant viruses is a very interesting yet underexplored topic, especially given their much larger gene repertoires than those of most viruses. Overall it is intriguing to note differences between polar and nonpolar viromes and their gene repertoires. However, this manuscript seems to have relatively little discussion of the results and their causes and implications.

My major comments are as follows:

1. The authors emphasize the concept of viral adaptation to a specific environment and distinguishes it from viral adaptation to specific hosts. But viruses only express their genes inside host cells, where genomes with different fitness can be naturally selected. It is therefore hard to imagine that viruses adapt directly to a specific environment. Similarly, the temperature optima estimated for viruses should also, to a large extent, depend on the temperature optima of their hosts.
2. One uncertainty in the analyses is that MAGs were assembled from DNA reads from a mixture of microbes. Some of the virus MAGs might contain sequences of host origin due to misassembly. Some others might be GEVEs plus some neighboring host sequences that were not really part of viral genomes.
3. The geographic distribution of the eukaryotic and giant virus MAGs can greatly influence the distribution patterns of gene repertoires. The distribution maps should be included in the main figures (and discussed in main text) and tests should be done to show that the polar/nonpolar barrier is not observed due to biased distribution of sampling sites. Also, there seems to be very few samples from the Southern Ocean (Extended Figure 1), so maybe this study is more representative of Arctic instead of all polar regions.
4. The authors pointed out that the divergence of giant viruses predated the divergence of eukaryotes (line 200). In that case, the host analyses should also include prokaryotes instead of assuming these viruses only infect eukaryotes.

Minor comments:

1. line 178: Here *Emiliana huxleyi* is used as an example of polar-distributed species, but it seems that it is cosmopolitan in its distribution (including tropical regions). Also many of their viruses seem to have been isolated from nonpolar regions.
2. line 70: Ref 27 doesn't appear to say much about the overall dynamics of losing and gaining genes in giant viruses. Probably some other studies focused on gene repertoire evolution should be included here.
3. line 200: Does Ref 23 provide any evidence to show that the divergence of giant viruses predates the divergence of eukaryotes?
4. line 285: Like *Emiliana huxleyi*, *Cafeteria roenbergensis* also seems to have a widespread distribution instead of being polar.
5. line 315-316: Is there any proof of concept that the method can really detect polar-adapted genes in eukaryotes?
6. line 321-323: "The result further suggests that virus-host horizontal gene transfer is not the primary driver of viral polar adaptation, and that genomic adaptations are uncoupled between viruses and eukaryotes." If it is not through horizontal gene transfer or during infection of hosts, how could these viruses have obtained these genes in the first place?

Reviewer #1 (Remarks to the Author):

This research article provided an insightful analysis of the divide between polar and nonpolar giant viruses (NCLDVs) in the Tara Oceans dataset. Through a variety of bioinformatic approaches, it was shown clearly that this division exists and there are several adaptations that polar viruses may have to inhabit this area. The enrichment analysis of KO groups, **something commonly done in human RNAseq analysis**, was a good approach to show differences and a highlight of the article. The classification of ecological optimum and polar vs non-polar viruses through taxonomy and abundance was also a novel approach that deserves praise. **There do seem to be some issues with assumptions made when comparing RPKM across samples on top of some minor issues that need to be taken care of.** Overall, this study pushes forward our knowledge of understudied polar viruses and sets forward a methodology that could be used for further classifications and analyses.

We are grateful to the reviewer for taking the time to review our manuscript and for your positive feedback. We appreciate your recognition of our novel approach and its potential for advancing the study of understudied polar viruses. We largely agree with the constructive comments made by the reviewer.

My comments and suggestions are provided below:

General introduction: The authors switch between talking about virus adaptations and virus-host adaptations which seem like two different things. The authors should focus on one of these two categories or talk about them in separate paragraphs.

We apologize for the confusion caused by our narrative. You're correct that we should focus on our category: our primary focus in this study is on virus adaptation to their environments, not the virus-host adaptation. In fact, virus adaptation is not an unexpected phenomenon, for example, cyanophages have been shown to adapt to low phosphorous environments by acquiring phosphorous-assimilation genes (From Kelly et. al. 2013. <https://pubmed.ncbi.nlm.nih.gov/23657361>).

We've cited this work in the revised manuscript to clearly introduce the notion of virus adaptation. Corresponding part of the text now reads as follows:

(Ln 51-53)---*“Some viruses are known to adapt to environments by acquiring specific metabolic genes. A notable example is cyanophages in low phosphorous environments, which tend to possess phosphorus assimilation genes.”*---

We also introduced a new discussion in Ln 355-368 to clarify our standpoint, which reads ----*“For a virus to adapt to a new environment, it is a prerequisite that its host already adapts to the environment. This host adaptation may induce additional “environmental” changes for the virus besides the changes in physico-chemical extracellular conditions. Virus-host interactions involve different processes such as adhesion to the cell, metabolic remodeling, viral genome replication, genome packaging and egress from the cell. These processes are likely affected not only by physico-chemical conditions (such as temperature) but also, and more profoundly, by the biochemical and physiological conditions of the host cell. Therefore, the adaption of a virus to “environmental” and “environment-induced host cellular conditions” may be manifested in viral genomic changes. Such viral adaptation to the “environment” may require alteration*

or innovation of viral metabolic strategies, which may be independent from the host strategy in their adaption in the same habitat.”---

We appreciate your comment because it has helped us to substantially improve the clarity of our manuscript.

Ln 39-44: This introduction seems overly broad and certain word choices are questionable such as “lush” and “nourish”. I also think showing polar bear adaptations is a bit irrelevant in a paper about viruses.

We consider it is a matter of opinion. According to your suggestion, we discarded the word “lush”, while we kept the use of “nourish” in the revised version of our manuscript.

We have improved the language here:

(Ln 40-42)---“Nevertheless, high primary productivity of phytoplankton in these regions nourishes a diverse range of creatures, ranging from microscopic organisms to large animals.”---

Introducing the animal case (polar bear) is our intention to draw attention to the high primary productivity of the polar regions and to the generality (from microbes to animals) of the adaptation to low temperature environments regardless of the differences in the underlying genetic changes. Therefore, we kept this example.

Ln: 54: The sentence “A large proportion of Arctic-specific genes from these viruses were suggested under positive selection based on their mutation patterns.” needs to be reworked as I am not sure what it is saying. The next sentence also could use rework and a citation of where the information is coming from.

The pattern, to be specific, referred to the high ratio of non-synonymous to synonymous mutation rates (i.e., pN/pS) for those genes.

Now we modified the corresponding sentence as follow:

(Ln 56-58)---“A substantial proportion of genes specific to these viruses were suggested to be under positive selection based on the ratio of non-synonymous to synonymous mutation rates.”---

This result comes from Gregory et. al. 2019, <https://pubmed.ncbi.nlm.nih.gov/31031001/>, which was now cited in the sentence right before this description (Ln 56).

Ln 54-62: The wording used seems to imply that it is the viruses themselves adapting to the environmental stressors when most of the papers you cite refer to the virus-host system as the unit doing the adapting. It might be useful to elaborate a bit on how viruses themselves are adapting - as the adaptation of viruses is likely a direct consequence of the adaptation of their hosts?

Thank you again for pointing this out. We agree that our previous texts were unclear about the distinction between viral adaptation and virus-host adaptation. According to your comments, we extensively edited the corresponding paragraph in the introduction (Ln 355-368), and added a new discussion as mentioned above in our response to your first comment.

Ln 76: Mirusviruses are brought up without any introduction. It would be helpful to introduce them and why you are focusing on them specifically.

We apologize for the incomplete initial introduction of Mirusviruses. The oversight was mainly due to a parallel project (Gaia et. al. 2023, <https://www.nature.com/articles/s41586-023-05962-4>), which provides more extensive details on Mirusviruses. We were trying to avoid duplicate publication.

We have now enhanced our introduction to Mirusviruses. The revised text is as follows: (Ln 78-80)---*“Mirusviruses are large dsDNA viruses, which widely distribute in the global ocean, likely infect marine plankton, and share large and similar gene repertoire with viruses of the Nucleocytoviricota.”*---

Ln 87-92: A bit more detail on the GOEV will be helpful here, although the authors included the citation. For example, authors should include how many of the genomes in these dataset were assembled from Tara Oceans dataset. If most genomes in GOEV were assembled from Tara Oceans data in the first place, it is expected that these genomes will have coverage from at least one Tara Oceans sequence library.

Same as the previous comment, we have been doing the parallel projects. The GOEV was developed and the details were initially published in the paper (Gaia et. al. 2023, <https://www.nature.com/articles/s41586-023-05962-4>).

We aimed to avoid redundancy and potential self-plagiarism by not repeating extensive details from the GOEV paper. Therefore, we provided a concise summary of the GOEV database in the **Supplementary Text** of our previous manuscript:

(ST Ln 13-17)---*“The initial version of the GOEV database included 697 genomes reconstructed from 798 Tara Oceans metagenomes, 1,187 metagenome-assembled genomes (MAGs) from two previous metagenomic surveys, and 235 reference NCLDV genomes. We eliminated redundancy by implementing a cut-off with Average Nucleotide Identity (ANI) of 98%, ultimately resulting in a refined database containing 1,817 genomes.”*---

In responding to your comment, we now added this part to the Method section of the main manuscript (Ln 396-403).

---*“The initial version of the GOEV database included 697 genomes reconstructed from 798 Tara Oceans metagenomes, 1,187 metagenome-assembled genomes (MAGs) from two previous metagenomic surveys^{33,64}, and 235 reference NCLDV genomes. Redundancy in the dataset was reduced based on Average Nucleotide Identity (ANI) of 98% (by always retaining the 697 genomes from Tara Oceans metagenomes), ultimately resulting in a refined database containing 1,817 genomes.”*---

Regarding the coverage of the Tara Oceans sequence library, you are correct. All of the Tara-originated genomes (697/697) in our database have detectable coverage from at least one sample. This number could be collated in the **Supplementary Table 2** of this manuscript and Gaia et. al. 2023.

Ln 97: Do you mean R²? If not, the author may want to explain what R-value is.

The R statistic in the ANOSIM test represents the rank dissimilarity between and within groups. It provides a measure of the degree of separation between groups of samples.

The value of R ranges from 0 to 1. An R-value closer to 1 would indicate strong group separation, supporting a strong effect of your grouping variable on the community composition. In the revised version, the corresponding text reads as follows:

(Ln 101-103)---*“The R value of ANOSIM test (intergroup dissimilarity) increased from 0.4021 to 0.6141 after merging three nonpolar biomes, demonstrating the existence of a clear polar barrier for giant virus communities.”*---

This finding reveals that the viral community within the polar biome displays the highest dissimilarity when compared to the pooled communities of the three other biomes (Coastal, Trades, and Westerlies), supporting our claim for the existence of the polar boundary.

The original paper about ANOSIM has been cited:

(Clarke. 1993. <https://onlinelibrary.wiley.com/doi/10.1111/j.1442-9993.1993.tb00438.x>)

Ln 165: The author mentions that the relationship is unidentified but they just identified it. Also, given the biology of most diatoms (the silica shell), and lack of any direct evidence of host virus interaction in culture or genomic signatures, I would be cautious about inferring diatom-NCLDV host associations. The authors might want to add a **caveat** here.

Your suggestion is certainly valid. We have carefully restructured the paragraph and attempted to temper the claims made on virus-host prediction.

(Ln 179-181)---*“However, this in silico prediction is limited by the absence of direct evidence of host–virus interaction and by the current availability of isolated Polar Chaetocerotales genomes.”*---

Ln 178-180: this argument seems circular as viruses were assigned to categories based I’m assuming in part on isolation location. Validating this metric would require new viruses that weren’t part of your training dataset. It seems self-evident that a virus only isolated in Arctic oceans would be classified as polar. I might be wrong in my interpretation, so please provide further discussion to address this concern.

Thank you for highlighting this crucial point. It is true that the example of Proculviricetes (detected in Tara polar samples) was circular. Therefore, we separate this sentence from the validation part. This Proculviricetes part now reads as follows in the revised version of our manuscript.

(Ln 189-191)---*“All six genomes of Proculviricetes, a recently discovered class-level group recovered exclusively from the Arctic and Southern Oceans, were classified as Polar viruses as expected.”*---

However, we believe that the control by viruses that were isolated/detected outside the Tara Oceans data can be used for validating our niche assignment method. Therefore, we retained the following sentences, after removing the example *Emiliana huxleyi*, which we now consider inappropriate according to a comment from Review #3:

(Ln 194-199)---*“However, several cases justify our niche assignment using global-scale abundance profiles. For example, Chrysochromulina ericina virus, isolated from high latitude Norwegian coastal waters, was correctly assigned to the Polar niche. The polar niche assignment to a metagenome-assembled genome (MAG) derived from Arctic samples was corroborated by its phylogenetic grouping with organic lake phycodnaviruses, which were independently derived from Antarctic organic lake metagenomes.”*---

Figure 3: The black circles around the circles are hard to see. I would make this bolder or come up with a different symbol to visualize it.

Thank you for the suggestion. We have updated the plot using a different symbol for P-value < 0.05 and P-value ≥ 0.05 .

Ln240: Again this seems circular to say that polar exclusive KOs are more abundant in Polar genomes if those genomes are where you most likely got the proteins from in the first place.

Thank you for pointing out this concern. Indeed, these polar-specific KOs are expected to be found in polar genomes as you mention. However, the key point here was their proportion in individual genomes, which cannot be predicted without analyses. In fact, the proportion was 15.84%, which was notably large for us. Therefore, we described these proportions. To emphasize this narrative, we rephrased the corresponding text as follows in the revised version.

(Ln 266-268)---*“The average proportion of polar-specific KOs among all genes with KO annotations in a viral genome was 15.84% for Polar genomes, which was significantly higher than Nonpolar (6.95%) and Unknown (7.93%) genomes.”*---

Ln 256-267: This paragraph could benefit from more flow between sentences as sentence structure is quite repetitive.

Thank you, we have rearranged the paragraph.

(Ln 282-293)---*“At the pathway level, three pathways were significantly enriched with Polar-specific KOs (Fig. 3b; Fisher’s exact test, $P < 0.05$). These were unsaturated fatty acid biosynthesis, N-glycan biosynthesis, and cholinergic synapse pathways. A high proportion of unsaturated fatty acids is known to be an adaptive trait among bacteria inhabiting low temperature and high pressure environments⁵⁹. Eukaryotes and their viruses have similar membrane compositions to those of bacteria. Indeed, giant viruses isolated from high latitude areas encode enzymes for the biosynthesis of unsaturated fatty acids⁴⁴, which may be part of a strategy to rewire the host fatty acid physiology⁵⁶. N-glycan plays an important role in the virus replication cycle, including virus recognition and virus release, and potentially contributes to the stability of virions⁶⁰. The enrichment of Polar-specific KO in cholinergic synapse (albeit its pathway name reflects biology of animals and may not be relevant to unicellular eukaryotes) implies the ability of polar viruses to regulate signal transduction.”*

Ln 264: The most enriched should be mentioned first.

Thank you, this paragraph has been modified as above.

Ln 276: Seems self-explanatory like the above comments on Polar genomes showing lower temperature minima.

Yes, your observation is right that it seems self-explanatory. However, this sanity check is necessary because Fisher's exact test, which we used to detect overrepresentation of KOs in polar viruses within specific lineages (from root to genus levels), was performed within a subset of the entire tree. Namely, every single test does not look at the distribution of KOs outside the clade being tested. In such a context, a tendency detected locally (in a subtree) could potentially contradict the global tendency. For instance, a KO that is enriched in polar genomes within a subtree (Fisher's exact test) might be specific to tropical waters globally (Optimum). This is why the sanity check was necessary. Our

results (see Extended Fig. 9c) indicate that KOs enriched in the polar genomes (within clades) did not exhibit such an unfavorable inconsistency.

Ln 283-287: Mentioning bacteria here is odd as these viruses infect eukaryotes. This seems like a stretch to claim they are surviving by coating their capsid with these bacteria-like proteins without in-situ data. This is a strong claim to make, and I would be cautious about such a general assumption.

We appreciate your valid concern, and we agree with you. We now modified the corresponding paragraph as follows (Ln 312-316). --- *“Enzymes of CMP-KDO biosynthesis were previously found encoded by a giant virus and suggested to add glycoconjugates to the surface of virions to enhance virion-cell recognition. The genomes in the examined Polar clade may also coat virions with glycoconjugates, to potentially enhance interactions with their polar hosts and/or virion stability.”*---

Ln 326: This is not a good hook sentence for the conclusory remarks. Functional repertoire doesn't feel like a “trait”.

We agreed and revised the sentence as:

(Ln 371-372)---*“The adaptation of cellular organisms to their environments could be largely manifested in their functional repertoire.”*---

Ln 336: I would change presently to “previously” as you characterized it in this study. Thank you for the suggestion, but we didn't use “previously unidentified” because it implies “we have identified now” (which is not the case). The diatom part is all about in silico without experimental evidence. The revised sentence now reads: (Ln 381-383)---*“Consistent with these findings, our analyses implied some potential polar host, such as diatoms, could contribute to the polar distribution of giant viruses.”*--- This modification also takes into account your other comments regarding Chaetoceros.

Ln 345: It seems like a stretch to make claims about climate change here as it wasn't really the focus of any of your discussion or introductory material.

We have deleted this part. Thank you for your precious comment.

Ln 359: I might be wrong, but as far as I understand, mean coverage wasn't what was converted, raw read counts were.

Indeed, both RPKM and mean coverage in the manuscript are derived from raw read counts. However, mean coverage, while normalized to genome length, does not take the difference in the sequencing depth of samples into account. In our study, we utilized both metrics, of which RPKM being further normalized from the mean coverage by the sequencing depth of metagenomes (read depth is available at the EBI under project PRJEB402). This normalization allows RPKM to provide a reasonable comparison across samples.

We recognize that our initial explanation might have caused confusion, and for that, we extend our apologies. We have now shortened the description in the methodology section to eliminate any obscurity and provide a clearer understanding of the metric used.

(Ln 411-412)---*“Relative abundance of a giant virus in each sample was calculated in Reads Per Kilobase per Million mapped reads (RPKM).”*

Ln 360-362: It would seem that comparing RPKM across all 900+ samples is a rather

'bold' approach as RPKM can vary greatly if total DNA is different as well as sample distribution

(<https://www.ncbi.nlm.nih.gov/pmc/articles/PMC7373998/#:~:text=RPKM and TPM represent relative,are close to each other.>).

While I understand that this normalization was perhaps necessary, it can result in misleading conclusions in my opinion. RPKM works best when the result is compared within the sample (i.e., which virus is most abundant within a sample compared to others?) - but comparing RPKM across such a large number of samples across this vast ecosystem can be tricky. The authors should discuss potential caveats of using RPKM and how that could potentially bias the interpretation of the results. It is also possible that I am missing something obvious, but further discussion should clarify these issues. For example, maybe RPKM is fine for 'correlation' purposes?

We fully appreciate the importance and complexity of using normalization metrics such as RPKM, TPM, and others in cross-sample metagenomics analyses. We agree that they can each present unique strengths and potential challenges when applied across diverse ecosystems. In our study, we chose to use RPKM (employing the same equation as in the paper you referenced) for the following two reasons:

- 1) We compared mean coverage and RPKM, and found no substantial difference in the results of our niche assignment and function enrichment analyses.

$$\text{TPM} = 10^6 * \frac{\text{RPKM}}{\text{Sum(RPKM)}}$$

- 2) TPM, which mainly differs in that it uses the total abundance of NCLDV as the denominator, may be the best metric for RNAseq. However, we believe it might not be the optimal choice for environmental samples. In environment samples, the absolute abundance of NCLDVs is typically low (lower than bacteria). Using the cumulated abundance of NCLDVs in one sample as the denominator (as TPM) could potentially overstate the relative abundance of NCLDVs in communities where NCLDVs are in fact rare.

Meanwhile, there are some peer-reviewed papers that used RPKM (i.e. sequencing depth-normalized metric):

---"*For the macrodiversity calculations, the average read depth was used as a proxy for abundance and normalized by total read number per metagenome to allow for sample-to-sample comparison.*"---

(Gregory et. al. 2019, <https://pubmed.ncbi.nlm.nih.gov/31031001/>)

---"*Relative abundance of a phage in each sample was calculated in RPKM.*"---

(Weinheimer et. al. 2022, <https://pubmed.ncbi.nlm.nih.gov/35260829/>)

Also in the metagtranscriptome paper :

---"*Unigene expression values and genomic occurrences were computed in RPKM (reads per kilo base covered per million of mapped reads).*"---

(Carradec et. al. 2018, <https://pubmed.ncbi.nlm.nih.gov/29371626/>)

Additionally, our application of this method, as demonstrated by positive controls such as OLPV and CEV, shows that RPKM-based calculations can appropriately assign niches to clades in viral phylogeny. We believe that its benefits and drawbacks considered, RPKM serves as a suitable metric in our study context.

We have updated the discussion section with a potential issue of using RPKM.

(Ln 193-194)---“Limitations, such as unequal sampling and sequencing depth, may potentially affect niche assignment.”---

Ln 369: What does “tree structure manipulation and analysis” mean? The author should explain what was done here for the sake of reproducible research.

This was an over-literal way to describe pruning, labeling, scaling and other standard procedures for visualizing phylogenetic trees. Specific analytical manipulations, such as rooting, are mentioned where relevant in the Results and Discussion section.

We improved the corresponding method text:

(Ln 421-422)---“Tree visualization and analysis were carried out using ETE3 toolkit v.3.1.1”---

Ln 446-450: This section should be explained further as it is confusing why this was done. Previous methodology should be cited as to why it was calculated this way.

This method was designed in the present study. We have added an additional explanation about this method:

(Ln 493-495) ---“The size index, corresponding to the filtration fraction, was designed in this study and serves as an indicator of the host range, with a larger size index implying that the virus infects larger-bodied organisms.”---

The results were shown in the **Supplementary Text**.

Ln 456-457: This part needs further explanation in terms of the methods.

The method has been explained as:

(Ln 506-512) ---“First, we assigned genomes with zero mapping signals from “Nonpolar” metagenomes to “Polar” biome niche. Likewise, we assigned genomes with zero mapping signals from “Polar” metagenomes to “Nonpolar” biome niche. Additionally, utilizing RPKM profiles, we determined the statistical significance to assess whether a specific genome is overrepresented in either “Polar” or “Nonpolar” metagenomes using the Wilcoxon rank-sum test. The Benjamini-Hochberg (BH) correction was applied to significance values to account for the effect of multiple testing.”---

Ln 467: See the above comment about RPKM but I do not think it is valid to compare it across samples without any normalization. This section could also be explained better as it seems critical to the thesis of the paper. I’m not sure if ecological optimum is biologically relevant to viruses and if so, a paper should be cited to back up why you are calculating it. Otherwise, the authors should further discuss the relevance of this metric in the context of their research and discuss if it is a novel concept in regards to virus ecology.

Thank you for your comment. It's worth emphasizing that our use of RPKM is deliberate and not without careful consideration. Please refer to our detailed explanation provided earlier regarding the use of RPKM across multiple samples.

Additionally, to substantiate our choice, we now compared temperature optima calculated using two methods (mean coverage vs. RPKM):

The comparison clearly indicates that both RPKM and mean coverage portray a similar pattern of polar/nonpolar distribution pattern for viral genomes, as represented by the plotted dots. This distribution pattern is our most critical research objective, which is notably similar between the two methods, even though the mean coverage method doesn't normalize sequencing depth.

Furthermore, we can see from the plot that the RPKM-based method, which considers the impact of sequencing depth, conservatively estimates the fraction of polar genomes. We believe our RPKM-based calculation has provided evidence to reflect the reality of global NCLDV biogeography.

Your suggestion to elaborate on the concept of the ecological optimum is well received. In ecology, the ecological optimum represents the conditions under which a species is most likely to thrive, considering various environmental factors.

Our approach is not entirely novel but has been employed in previous plankton studies: (Chaffron et. al. 2021, <https://pubmed.ncbi.nlm.nih.gov/34452910/>)

Overall, we agree that a more discussion on the relevance of this metric could enhance the readability and reliability.

Therefore, we added a short description as follow (Ln 522-524): --- *“Given the polar barrier for giant viruses, which was described previously (Endo et, al, 2020) and confirmed in this study, we considered this metric is also useful to characterize ecological properties of viruses studied here.”* ---

We have also revised the method to clarify the technical aspect of the ecological optimum.

(Ln 526-529) ---*“For example, if the proportion of RPKM for Genome1 in sample1 represents 5% of the Genome1’s cumulated RPKM across all samples, then 5% of the elements of the weighted vector will be filled with the environmental value measured for sample1 (e.g., temperature and latitude of sample1).”*

We appreciate your insightful feedback in this regard.

The author brings up Mirusviruses in the abstract and briefly in the introduction but

then doesn't talk about them at all in the analysis. This group should either be talked about or removed from the study.

We agree with your suggestion and have incorporated a brief result about the Mirusviruses into our results section.

(Ln 191-192) ---*"A lineage of mirusviruses (clade M2) formed a large clade mainly composed of Polar viruses with an additional sub-clade composed of Nonpolar viruses."* Specifically, we observe that Mirusviruses showed clear Polar and Nonpolar clades (Figure 2A), illustrating a case of the back-and-forth adaptive events between Polar and Nonpolar niches. We would like to also note that inclusion of Mirusviruses in our study contributed to our functional analyses. For instance, we detected several functions enriched in Mirus polar genomes, which can be found in Supplementary Table 4.

Again, we want to thank the reviewer for the very thoughtful comments, especially for the clarification of viral adaptation and virus-host system, as well as the precise usage of abundance normalization. We have modified our manuscript based on those comments. We believe the quality of the current version has been substantially improved.

Reviewer #2 (Remarks to the Author):

Meng et al present a metagenomic analysis of giant viruses in the ocean, with particular emphasis on comparing polar viruses to those found in lower latitudes. They perform a suite of computational analyses to demonstrate that giant virus communities are distinct at higher latitudes and that polar viruses encode unique genes and have undergone the transition to polar climates multiple times independently. Overall this is a comprehensive study that examines giant virus communities in polar marine communities that will interest a broad range of readers. The figures are compelling and the bioinformatic analyses are well done. In general I would recommend adding more details for some of the methods, and I have some specific questions regarding some of the informatics and a few other suggestions to potentially strengthen the manuscript.

We thank the reviewer for his thorough evaluation of our manuscript and his recognition of the study's comprehensive nature. We appreciate his positive remarks on our illustrations and bioinformatics analysis. In the revised version of the manuscript, we have detailed the methods and addressed specific queries and recommendations raised by the reviewer. We believe that these adjustments have enhanced the manuscript's robustness and comprehensibility to our readers.

Perhaps I missed it somehow but it is unclear to me how the niche assignments were made. On Line 487 it is stated that this was done as previously described, but no citation is given. More detail on this would be welcome because it is an important step that underlies many subsequent analyses.

We agree that the description was unclear. We've clarified this in our revised manuscript in two parts. First, we have modified the niche assignments method, "**Biome and size niche**", as:

(Ln 504-515)---*"Each sample was associated with one specific marine biome (Coastal, Trades, Westerlies, or Polar). To straightforwardly investigate the difference between polar and nonpolar regions, we labeled Coastal, Trades, and Westerlies samples as "Nonpolar". First, we assigned genomes with zero mapping signals from "Nonpolar" metagenomes to "Polar" biome niche. Likewise, we assigned genomes with zero mapping*

signals from “Polar” metagenomes to “Nonpolar” biome niche. Additionally, on the basis of RPKM profiles, we calculated the significance using the Wilcoxon rank-sum test. The Benjamini-Hochberg (BH) correction was applied to significance values to account for the effect of multiple testing. The significance threshold was set to a corrected P-value of 0.05. Similar assignments were performed for two size fractions: intercellular (Pico-size) and intracellular (Piconano, Nano, Micro, Macro, and Broad).”---

Regarding the section you pointed out, we have revised the sentence as follows to cite the method:

(Ln 548-549)---““Polar”, “Nonpolar”, or “Unknown” biome niche was assigned to each viral genome as described above (“**Biome and size niche**” section).”---

Line 29 - Can we be sure that temperature is driving these patterns, and not other features that are correlated with temperature? I.e. nutrient levels may be higher in polar oceans in the summer compared to lower latitude samples. In general this may be a point that is worth clarifying in other parts of the manuscript. In my experience many variables in marine datasets co-vary, so identifying the true drivers can be difficult. Thank you for the suggestion. Many variables indeed co-vary, making it challenging to disentangle the true drivers of observed latitudinal patterns.

In our study, besides temperature, we provided the environmental optima for a range of other environmental variables, including ChlorophyllA, NO2, PO4, Salinity, Si, and the community structure of eukaryotes in our previous manuscript (**Supplementary Table 2**).

In responding to the comment, we now performed a Spearman analysis to understand the correlation between latitude and different environmental optima for viral genomes. This analysis revealed that temperature was most highly correlated with the latitude (Spearman rho = -0.886), which was followed by salinity (rho=-0.432) and ChlorophyllA (rho=0.579). Therefore, temperature is the primary driving variable for the latitudinal distribution of studied viruses.

However, neither salinity nor Chlorophyll A appear to distinctly drive the virus-eukaryote network like Temperature does (Figure 1b in the revised manuscript).

It should also be noted that temperature is the most recurrently recognized important environmental parameter to explain the global distribution of marine microbial community (e.g. Sunagawa et al, Science, 2015, <https://pubmed.ncbi.nlm.nih.gov/25999513/>).

These additional analyses and argument were added in the revised version of **Supplementary Text:**

(ST Ln 59-67)---“*We estimated robust ecological optimum (temperature, salinity, latitude, ChlorophyllA, Si, NO2, PO4) for individual viral genomes (Supplementary Table 2). We performed a Spearman analysis to understand the correlation between latitude and the different environmental optima. Temperature was found most correlated with the latitude (Spearman rho = -0.886), which was followed by salinity (rho=-0.432) and ChlorophyllA (rho=0.579). No other variable had an absolute rho higher than 0.1. Moreover, neither salinity nor ChlorophyllA appear to distinctly drive the virus-eukaryote network like Temperature does (Figure 1b). Therefore, we considered temperature is the most appropriate variable to explain the latitudinal (i.e., Polar vs Nonpolar) distribution of viruses in this study..*”---

Line 66 - “active” - please consider citing <https://doi.org/10.1128/mSystems.00293-21>. I believe this is a clear demonstration of high giant virus activity in marine systems. Thank you for recommending this pertinent reference that indeed showcases the high activity of giant viruses in marine systems. We have cited this work in our revised manuscript.

Line 147 - it would be useful to know the percent of viral clades that have a predicted host according to this method. Right now it is hard to gauge how successful this approach was at predicting hosts. Also wouldn't host prediction occur at the genome level, and thus should be reported as such? E.g. XX percent of total MAGs had a predicted host according to this method. In general the paragraph starting on line 145 is a bit short, and I feel that more details could be added to improve readability.

Thank you for your constructive feedback.

Indeed, the Taxon Interaction Mapper (TIM) is a clade-to-clade host prediction method, designed to clarify associations derived from co-occurrence research. In total, 6.38% of total viral genomes (N = 88) had a predicted host according to this method.

We have added the percentage of the predictions.

(Ln 161-164)---“*This method links viruses and eukaryotes through a clade-to-clade relationship by testing whether leaves (i.e., viral genomes) under a node of the virus tree are enriched with a specific predicted host group. This method assigned predicted host*

taxa (five different taxa) to 34 viral clades, including 6.38% of total viral genomes (N = 88, Extended Fig. 7a)---

To address your concerns, we also improved the method section.

(Ln 476-480)---“If a specific eukaryotic group (NCBI taxonomies, including order, class, and phylum) was significantly enriched under a node of the viral tree, compared to the rest of the tree (determined using Fisher’s exact test and Benjamini–Hochberg adjustment), that specific eukaryotic group was deemed the predicted host.”---

The endogenous viral signals found in diatom genomes are quite interesting. I believe the writing of this section could be clarified so it is easier to understand whether the discussion is focusing on the endogenous PolBs recovered from the two previous studies cited, or whether this conclusion is primarily from the insertion signals identified here from the Delmont et al data. Moreover, the tree in ED Fig 7c is not entirely clear to me - the asfuvirales appear to cluster within the Imitervirales here, and both chaetoceros sequences are clustered together inside of that, so I’m not sure how a clear order-level designation can be made.

Thank you for pointing out the issue around the polB analysis. We have realized that previously used example, polB phylogenetic tree, was misleading and inappropriate. The Chaetoceros PolBs were detected by an independent HMMsearch with hits outside the viral regions (i.e. eukaryotic chromosomal regions). However, the detected polB seems to be more similar to eukaryotic homologs (thus probably not signals of an EVE).

Therefore, we decided to remove the PolB phylogeny from the results. However, we still keep the detection of viral regions in the diatom genomes. We have carefully modified the sentences in the section to avoid over-interpretation.

(Ln 178-181)---“So, the putative virus–Chaetocerotaes relationship may account for the diversity of giant viruses in high-latitude regions. However, this in silico prediction is limited by the absence of direct evidence of host–virus interaction and by the currently available genomes of polar Chaetocerotaes isolates.”---

To reinforce the viral signal detection analysis, we used ViralRecall to extract possible viral regions from 12 Tara Oceans Chaetoceros MAGs and 2 Chaetoceros isolates. To collect more potential viral regions, we used a parameter of -s 0 -w 15. Consistent with the ViralSorter results, multiple NCLDV-like regions (N = 198) were identified. (Each dot represents a detected viral region).

This new result was added to the revised version of our manuscript (Extended Fig. 7c).

Following is our new attempt to further strengthen the evidence for EVEs. However, we decided not to include this new analysis. It is presented only here for reviewing purpose.

We used Orthofinder to classify the ortholog groups of all proteins from giant virus genomes in the GOEV database. Then, an intra-main group protein alignment (Diamond blastP) was performed to establish the minimum similarity (bitscore) within the main groups for each OG.

Maingroup	OG0000000	OG0000001	OG0000002	OG0000003	OG0000006	OG0000026	OGXXX
Algavirales	101.3	650.2	234.2	297	759.2	625.5	XXX
Asfuvirales	228	758.4	607.1	898.3	578.2	2446.8	XXX
Imitervirales	80.9	272.3	210.7	97.4	392.9	507.3	XXX
Mirusviricota	127.5	341.7	0	243	973.8	740.3	XXX
Pandoravirales	188	503.1	346.3	308.5	963.4	813.1	XXX
Pimascovirales	125.2	347.4	251.9	314.7	418.7	550.4	XXX

Then we aligned the proteins from the above detected viral regions (12 MAGs + 2 isolate genomes) to all giant viral proteins in the GOEV database using Diamond. Only results with identity > 50% and coverage > 50% of the average length in an OG were remained. If a bit score of a viral region protein exceeded the intra-main group bit score, a potential “EVE-candidate” was labeled.

Using this approach, we identified 39 proteins from Chaetoceros viral regions that might represent a strong EVE signal, with 27 of them related to the Imitervirales clade specifically (red part in the pie-chart below). This result is consistent with the cooccurrence analysis.

The following are two examples (containing the highest number of proteins from Chaetoceros viral regions) wherein the labels marked in red represent sequences originating from Chaetoceros, and those marked in black are from viral genomes present in the GOEV database:

These results support our hypothesis that Chaetoceros are infected by NCLDV.

However, we decided not to add this new phylogeny result as this Chaetoceros-NCLDV relationship will still remain hypothetical and there would be many additional bioinformatics/phylogeny analyses required to make such analyses robust enough for publication. We considered that this level of in silico analysis is out of the scope of our current manuscript. We also noticed that Tara Ocean eukaryotic MAGs are not suitable for EVE studies, partly because when these MAGs were reconstructed, a manually supervised curation was performed that might have removed all viral-like fragments (especially contigs possessing NCLDV marker genes) from the final bins (<https://pubmed.ncbi.nlm.nih.gov/36778897/>).

Once again, thank you for pointing out this critical issue in the revision. Your contribution has greatly improved the quality of the paper.

Regarding the insertion signals found in the Delmont et al data, it would be useful to clarify what exactly these are in the main text, even though additional details are in the methods. Based on the methods it seems that mostly metagenome-derived eukaryotic

genomes from the Delmont et al study were used for this analysis, in which case it would be useful to confirm that the viral polBs are encoded on eukaryotic contigs (i.e. truly endogenized) and not viral contigs that may have been co-binned. Is it possible the viral PolB genes originate from viral contigs that were co-binned with the eukaryotic genomes? I understand that some details for this may be in the other paper, but given the importance of these claims it is worth providing details here. ViralRecall allows for contig-level plots to be generated (-f command), and so some contig-level VR plots showing the overall scores would be very useful here just to confirm they are chimeric (i.e. partly eukaryotic host, partly endogenized virus).

First, we checked the proportion of detected viral regions (VirSorter) in contigs. From the plot below, we can see most of the VirSorter detections are whole contig level, as you said. It is possible that the viral regions detected in MAGs may have been co-binned.

Please refer to the previous response. We prefer to entirely remove the phylogeny of endogenous element analysis and soften the claims about Chaetocerotales. Because we currently lack robust evidence supporting EVE, and the prediction of Chaetocerotales doesn't directly contribute to our central topic of polar adaptation. By doing so, we can keep the logic of the manuscript more on the virus polar adaptation, which is the primary focus of our study.

We appreciate your thoughtful feedback and the opportunity it has provided us to refine and focus our research.

Figure 2- has this tree been rooted at the Asfuvirales? It is somewhat difficult to see so I would just like to confirm. The RED index may yield misleading results depending on the root placement.

Thank you for your question regarding the rooting of our tree in Figure 2. We confirm that the tree was indeed rooted at the Asfuvirales. In addition, Chitovirales (in Pokkesviricetes), which are not well represented in marine datasets were not included.

We fully acknowledge the significance of root placement for an accurate interpretation of the RED index. Indeed, our approach to phylogenomics and marker gene usage slightly varies from that used in the cited paper: Aylward et. al. 2021, <https://journals.plos.org/plosbiology/article/authors?id=10.1371/journal.pbio.3001430> Nevertheless, we used the most informative genes (polB, RNAPa, RNAPb, TFIIS) for our phylogenomic analysis. These genes are crucial in elucidating the evolution of giant viruses. Furthermore, we rooted the tree at the Asfuvirales (part of Pokkesviricetes) in a similar way as Aylward et al 2021 (which used Pokkesviricetes as the outgroup). So, the derived RED values in our analysis remain comparable to the paper given above. We think our conclusions based on RED values are not affected by the possible small differences in the RED values.

To address your concern, we added a plot of RED values of viral taxonomic groups. Now we have improved the legend of Fig2 c,d.

(Fig2) ---“c, Relative Evolutionary Divergence (RED) values for viral orders and families. *N* stands for the phylum Nucleocyotviricota and *M* stands for Mirusviricota. **d**, Histograms of RED values for the nodes at which “polar” or “nonpolar” adaptation events were inferred. RED values of child nodes in adaptation events were shown.”---

Line 377 - when calculating alpha diversity, were the reads from the metagenomes sub-sampled? The different TARA stations have different sampling depths and this drastically impacts the alpha diversity estimates, so read rarefaction is needed for comparative purposes.

Thank you for your insightful query regarding the calculation of alpha diversity. First, it's important to clarify that the metagenomes we used did not significantly differ in terms of order of magnitude.

Moreover, there was no substantial difference in the number of reads among the samples of different biomes, especially between polar and nonpolar biomes. In our results, we observed a consistent size-dependent latitudinal pattern (Extended Fig. 1a, left panel), irrespective of whether polar samples had relatively more or fewer reads. Thus, neither the latitudinal pattern nor the boundary between polar and nonpolar regions was significantly influenced by this sequencing-depth issue. Furthermore, the Shannon index is less sensitive to sampling depth compared to other metrics such as species richness.

We added an analysis to evaluate the influence of including smaller RPKMs, which could be excluded after subsampling. Specifically, we assigned a value of '0' to the lowest 0.5%, 1%, 5%, 10%, and 50% of the total number of non-zero RPKM values. Samples with a larger sequencing depth normalize viral abundance to RPKM with a larger

denominator (total number of reads). So deeper sequencing metagenomes could yield smaller RPKM values, which are more likely to be discarded during the trimming process. The trimming process was performed for six size fractions, independently. As shown below, while the overall Shannon diversity value decreased as more RPKM values were discarded, there was no substantial difference in the latitudinal pattern for each size fraction. These results also support that the Shannon index is stable even when the data is rarefied.

Thank you for the concern. We have added the result above into the Extended Figure 5 in our new manuscript.

For the functional enrichment analysis, it would be good to see a supplementary spreadsheet with each KO and KEGG together with the counts used for statistics and the p-values. I couldn't find this- apologies if I missed it.

Thank you for your suggestion. We appreciate your interest in the details of our study. In **Supplementary table 4**, we now have included a supplementary spreadsheet detailing each KO and KEGG pathway along with the corresponding counts used for statistics and p-values.

Figure 3b - I am not sure what the size of the bubbles indicates - the scale is shown but the units are absent.

Thank you for pointing out this omission in Figure 3b. The size of the bubbles represents the number of KO detected in Tara Oceans, but we mistakenly left off the units from the scale. We have rearranged Figure 3.

Regarding the discussion on lines 256-267, is it possible that some pathways are significant due to few or individual KO enrichment, rather than presence of the entire pathway? I have noticed that with pathway-specific enrichment I often get false-positives because a small number of pathway components are present in high abundance, even though the others are missing. For example, a divergent acyl coa reductase may be highly abundant and make all of fatty acid metabolism look enriched, which would be misleading. For this reason I usually perform KO-specific enrichment and then look to see if the enriched KOs tend to belong to the same pathway. For example, are other genes in sphingolipid metabolism enriched in polar MAGs, or just the sphingolipid glycosyltransferase? Sphingolipids are likely involved in apoptosis stimulation as a means of viral egress (probably worth citing Vardi et al Science here [10.1126/science.1177322](https://doi.org/10.1126/science.1177322)), so it would be interesting to know if this was a common strategy for egress in polar viruses.

Thank you for your insightful comment. We understand that the presence of a few or individual highly abundant KOs could potentially skew the perception of an entire pathway being enriched depending on the detection method. However, our approach is less biased by such effects. For a given pathway, we measured the fraction of components (i.e., enzymes) that were defined as Polar-specific KOs. This fraction was tested by Fisher's exact test. Such our approach is different from other methods that use quantitative data (such as RPKM). Our method was not well described in the previous manuscript. Therefore, we decided to add an explanation in the Method section as follows:

(Ln 555-561): ---*“Polar-specific KOs were defined as those with a temperature optimum below 10°C and a latitude optimum above 50°. For pathways with at least half of the detected KOs as polar-specific KOs, we compared the fraction of components (i.e., enzymes) defined as polar-specific KOs with the fraction of all other pathways. This fraction was tested by the Fisher's exact test and adjusted by the Benjamini-Hochberg (BH). This analysis excluded rare KOs (observed in fewer than 5 genomes). To avoid the enrichment of pathways led by sparse KOs, the pathways were exhibited only if the number of detected viral KOs in a given pathway constituted more than 10% of the total number of KOs in the pathway.”*---

In the case of sphingolipid metabolism, we found that only the glycosyltransferase was uniquely encoded in polar MAGs, but no enrichment was observed at the pathway level. Therefore, this enzyme was mentioned in the KO-level analysis part. This example highlights the value of employing different metrics (KO-level, pathway-level) in our analysis.

I feel like the presence of the “neuroactive ligand receptor interaction” is a bit strange and possibly a false-positive, perhaps because some ankyrin repeat proteins have spurious matches to proteins in the KEGG database. Manual annotation of some of the proteins that are driving this pattern could reveal more detail and confirm this observation.

Basically, there are two types of proteins in neuroactive ligand receptor interaction. One is trypsin, which is widespread in Chlorophyta genomes. Therefore, the presence of this gene in viruses is not strange. The other one is nicotinic acetylcholine receptor, which is usually encoded by animals. In our database, seven of the viral genomes contained

nicotinic acetylcholine receptor encoding genes. Of these, six were assigned to Polar niche. Accordingly, this KO was also considered as a Polar KO.

The neuroactive ligand receptor interaction might be applicable to higher animals, but the corresponding pathways in lower plankton might require different annotations. We have regenerated the enrichment test result by setting a cut-off of “*the pathways were exhibited only if the number of detected viral KOs in a given pathway constituted more than 10% of the total number of KOs in the pathway.*” to avoid spurious enrichment in large pathways that only contain sparse viral KOs, like neuroactive ligand receptor interaction. However, the problem still exists on the cholinergic synapse pathway. So, we have improved our description in the revised manuscript as:

(Ln 291-293) ---“*The enrichment of Polar-specific KO in cholinergic synapse (albeit its pathway name reflects biology of animals and may not be relevant to unicellular eukaryotes) implies the ability of polar viruses to regulate signal transduction.*”---

Line 287 - “to enhance their interactions with Polar hosts.” - is it possible it also plays a role in virion stability?

Yes, that seems plausible! We have added this to the discussion.

(Ln 289-291) ---“*N-glycan plays an important role in the virus replication cycle, including virus recognition and virus release, and potentially contributes to the stability of virions⁶⁰.*”---

(Ln 314-316) ---“*The genomes in the examined Polar clade may also coat virions with glycoconjugates, to potentially enhance interactions with their polar hosts and/or virion stability.*”---

Would it be possible to make a table of the most enriched KOs in polar viruses? It would be useful to present a table of the top 10 or 20 most enriched genes. Given the emphasis on viral genomic adaptations I feel that the KEGG analysis may deserve more weight. Thank you for your suggestion. A supplementary spreadsheet has been added. The top 20 most enriched KOs in each level (overall, main group, family, genus) have been summarized into a new spreadsheet in the **supplementary table 4**.

It was difficult for me to interpret the supplementary spreadsheets because the column headers were not described. I usually include a tab in the spreadsheet that includes descriptions of the column headers, and that may be useful here. For Supplementary Table 4 I could not see p-values for the statistics, and I was hoping to find the 314 polar-specific genes mentioned in the text (line 212).

Thank you for pointing out the need for more detailed column headers in our supplementary spreadsheets. We have improved all the supplementary tables. We also added p-values in **supplementary table 4**.

I don't quite understand the difference between ED Fig 1d and ED fig 5. Some additional explanation in the legends would be welcome. In ED 5c it appears the diversity of Imitervirales increases markedly at higher latitudes, but this is not the case in ED Fig 1d. Is there a mid-latitude peak in giant virus diversity? It would be very interesting if there was a bimodal diversity peak.

Thank you for drawing attention to this matter. We regret any confusion caused by the differences between Extended Figure 1d and Extended Figure 5.

The difference that you pointed out for Imitervirales is due to the difference of the size fractions analyzed: ED Fig1d (right panel) is for “small size fraction (Pico, Piconano, Broad)”, while ED Fig5c is for “large-size fraction (Nano, Micro, Macro).

These were explained in the respective legends, but we added further detail. We modified the label for the x-axis of the right panel of ED Fig.1d.

Extended Fig.1 legend (last part): --- *“Locally estimated scatterplot smoothing plots of the latitudinal distributions of viral diversity (Shannon's index). The left panel presents the total diversity of all giant viruses along a latitudinal gradient in different size fractions. The Broad and Piconano size fractions were pooled because of their similar relative abundances and lack of Arctic samples in the Piconano size fraction. The right panel shows the diversity of communities of six main groups in the small-size fractions (Pico: 0.2–3 μm , Piconano: 0.8–5 μm , and Broad size: 0.8–2000 μm fractions).”*---

Extended Fig.5 legend: --- **“d**, Shannon's index of communities of six main groups in large-size fractions (Nano: 5-20 μm , Micro: 20-200 μm , and Macro: 200–2000 μm size fractions).”----

The latitudinal gradient distribution patterns are clearly distinct in small and large-size fractions. In particular, the distribution pattern of small-size fractions, especially within the broad (0.8–2000 μm) size fraction, seems to exhibit bimodal diversity peaks. These varied distribution patterns could potentially be explained by the differing host ranges of viruses in various size fractions. However, this observation is restricted by the unevenness of the sampling, such as the lack of samples from the Southern Ocean. We appreciate your insightful suggestion, and have revised the manuscript accordingly.

(Ln 150-151) --- *“We observed a mid-latitude peak in small-size fractions, and hotspots of viral diversity in the Arctic regions within large-size fractions.”*---

Review written by Frank Aylward

We appreciate the insightful comments from Prof. Aylward. These inputs, especially for the EVE part as well as the precise usage of abundance normalization, have significantly helped us to resolve fundamental issues and standardize the manuscript's format. We've revised our manuscript based on these comments and firmly believe that the quality of the current version has been markedly enhanced.

Reviewer #3 (Remarks to the Author):

Meng et al. analyzed the gene repertoires of giant virus genomes, including metagenome-assembled genomes (MAGs) from global oceans, and inferred the existence of an ecological barrier between polar and nonpolar giant virus communities, with the former having genes/functions that indicate their „genomic adaptation“ („adaptation by alteration of gene repertoire“) to polar environments in a way that is independent of their host adaptation. The adaptation of marine giant viruses is a very interesting yet underexplored topic, especially given their much larger gene repertoires than those of most viruses. Overall it is intriguing to note differences between polar and nonpolar viromes and their gene repertoires. However, this manuscript seems to have relatively little discussion of the results and their causes and implications.

We are grateful for your comprehensive review of our manuscript, and for acknowledging the novelty of our work on the genomic adaptation of marine giant

viruses. We understand your concerns regarding the need for more discussion on our results and their implications. In response to this, we have expanded our discussion in the revised manuscript.

My major comments are as follows:

1. The authors emphasize the concept of viral adaptation to a specific environment and distinguishes it from viral adaptation to specific hosts. But viruses only express their genes inside host cells, where genomes with different fitness can be naturally selected. It is therefore hard to imagine that viruses adapt directly to a specific environment. Similarly, the temperature optima estimated for viruses should also, to a large extent, depend on the temperature optima of their hosts.

Thank you for your insightful comment. Indeed, similar comments were given by the reviewer #1. To clarify the “viral adaptation to environments” that we envisage, we introduced a new reference in the introduction. Furthermore, we added a new discussion. These changes are as follows:

Ln 51-53: ---“Some viruses are known to adapt to environments by acquiring specific metabolic genes. A notable example is cyanophages in low phosphorous environments, which tend to possess phosphorus assimilation genes¹¹.”

Ln 355-368: ---“For a virus to adapt to a new environment, it is a prerequisite that its host already adapts to the environment. This host adaptation would give rise to additional environmental (or micro-environmental) changes for the virus. Such micro-environmental changes include alterations of cell surface structures as well as intracellular metabolic states. Virus-host interactions involve different processes such as adhesion to the cell, metabolic remodeling, viral genome replication, genome packaging and egress from the cell. These processes are likely affected not only by ambient physico-chemical conditions (such as temperature) but also, and more profoundly, by the biochemical and physiological conditions of the host cell that adapts to the target environment. Therefore, for a virus to adapt to a new environment, it needs to cope with both environmental changes and environment-induced host cell alterations. Our results suggest that large and giant DNA virus adaptation to polar environments requires alteration or innovation of viral metabolic strategies, which is manifested in viral genomic changes. In this adaptive process, viruses appear to take their own strategies that are distinct from the host strategies for the adaption to the same environment.”---

For us, the most important part of this logic is that when a host adapts to an environment, it is no more the same organism that inhabited the original place (in terms of cellular and molecular structure and physiology). This notion seems to be often ignored (especially when we consider that human can live almost everywhere). We believe this environmental-change-induced host change is the main reason why viral lineages and genotypes alter across niche boundary.

With regards to the temperature optima of viruses, we concur that these likely depend on the temperature optima of their host organisms. However, determining the temperature optima of host organisms is a difficult task, given the lack of knowledge on viral hosts. In our study, we used the temperature optima of viruses just as a proxy to describe their distribution (and implicitly their host distribution) across different temperatures.

2. One uncertainty in the analyses is that MAGs were assembled from DNA reads from a mixture of microbes. Some of the virus MAGs might contain sequences of host origin due to misassembly. Some others might be GEVEs plus some neighboring host sequences that were not really part of viral genomes.

Thank you for your insightful comment regarding the potential for misassembly in the MAGs.

In our study, we have endeavored to minimize these issues by performing a manually supervised curation for viral MAGs binning (refer to: <https://anvio.org/blog/giant-viruses/> and <https://merenlab.org/2017/01/03/loki-the-link-archaea-eukaryota/>). This rigorous process should have significantly mitigated the issues you've highlighted. However, we concur that the potential for host origin sequences due to misassembly is a general challenge inherent to all binning results. We are confident that our method, underpinned by meticulous manual curation, is one of the most effective approaches to yield clean MAGs. Similarly, other viral MAGs utilized in our study that obtained from previous metagenomics surveys also went through stringent binning pipeline and quality-check methods. These studies can be referred to for more details. (Moniruzzaman et. al. 2020, <https://www.nature.com/articles/s41467-020-15507-2>; Schulz et. al. 2020, <https://www.nature.com/articles/s41586-020-1957-x>).

Additionally, we added a new analysis to verify whether the functions mentioned in the manuscript are encoded in real viral contigs. We examined all the contigs and added the information in two spreadsheets within **Supplementary Table 2**. One spreadsheet covers all the contigs, while the other specifically focuses on the contigs encoding the KOs mentioned in the manuscript (Ln 266-281). In the contigs encoding the mentioned KOs, an average of 13% and up to 38% of genes match best with viral sequences. Moreover, eight contigs encode at least one NCLDV marker gene. Specifically, we examined contigs encoding three CMP-KDO KOs from Fig.4. The taxonomy of the closest homologs was determined using eggno mapper. It is observable that most of the contigs encoding CMP-KDO functions have genes that match best with viral sequences (averaging 11% and up to 45%). Notably, several contigs encode hallmark genes of NCLDVs. And one contig encodes three NCLDV marker genes, including a MCP protein.

This result support that CMP-KDO KOs are likely from real viral contigs. We added this result as Fig.4 c in the new manuscript.

We understand the inherent challenges in assembling MAGs from a complex microbial mixture, and acknowledge the possibility of including sequences of host. So we have included the result above to support our findings in the new manuscript.

Ln 309-312: ---“A large proportion of other genes encoded in the contigs harboring the three CMP-KDO KOs best matched to viral genes (averaging 11% and up to 45%; Fig. 4c). Furthermore, four contigs encoded NCLDV marker genes. These data confirms that the identified CMP-KDO KOs are bona fide viral genes.”---

Please note that the GOEV database has been published and provides further details on our methodologies and findings.
(<https://doi.org/10.6084/m9.figshare.20284713>).

Thank you for your suggestion.

3. The geographic distribution of the eukaryotic and giant virus MAGs can greatly influence the distribution patterns of gene repertoires. The distribution maps should be included in the main figures (and discussed in main text) and tests should be done to show that the polar/nonpolar barrier is not observed due to biased distribution of sampling sites. Also, there seems to be very few samples from the Southern Ocean (Extended Figure 1), so maybe this study is more representative of Arctic instead of all polar regions.

Thank you for your insightful comment. We agree that the geographic distribution of eukaryotic and giant virus MAGs can greatly influence the distribution patterns of gene repertoires. We didn't choose to include distribution maps in the main figures in order to focus on the polar narrative. Based on your suggestion, we have added the map to the main figure 1 and a caveat related to this comment.

(Ln 193-194)---“Limitations, such as unequal sampling and sequencing depth, may potentially affect niche assignment.”---

Additionally, we performed a new analysis to check the issue caused by unbalanced sampling sites. Here we used dissimilarity to assess the existence of the polar boundary. First, we randomly selected 10 metagenomes from each 10° latitude interval (i.e., 0°-10°, 10°-20°...70°-80°). We then calculated the Jaccard dissimilarity of viral communities for all pairs of these 80 metagenomes.

The pairs were categorized into three groups: NonPolar-NonPolar, NonPolar-Polar, and Polar-Polar. Dissimilarities were plotted along the average latitude of each pairs or grouped for each pair group. Here, we restricted the result to pairs with latitude differences within 10 degrees. Particularly, a dissimilarity peak is observed in the pairs

of Polar-NonPolar samples with a latitude ranged from 55-60 °. This peak is significantly higher than those pairs in either the NonPolar-NonPolar or Polar-Polar pairs (Right panel, Kruskal-Wallis test). Furthermore, there were a clear lack of community pairs with low dissimilarities in the Polar-NonPolar comparison. This finding suggests the existence of an ecological barrier between polar and nonpolar regions, although potential sampling bias should still be considered.

Regarding the Arctic representative issue, while there is indeed a scarcity of samples from the Southern Ocean, we'd like to highlight (as seen in Extended Figure 8b) that two OLPVs (a virus detected in an Antarctic lake) fall within the same clade as Arctic MAGs (i.e., TARA_ARC_NCLDV_00083 and TARA_ARC_NCLDV_00053). Similar observations can be seen in multiple other polar clades including MAGs from both ARC (Arctic Ocean) and SOC (Southern Ocean), indicating the polar adaptation is to both Arctic and Antarctic. Here are two examples:

These findings suggest that the polar adaptation we observed is not solely representative of the Arctic, but also of the Southern Ocean. Nevertheless, we acknowledge the need for more extensive and balanced sampling from the Southern Ocean in future studies. We appreciate your valuable feedback and have discussed these points more thoroughly in the revised manuscript.

4. The authors pointed out that the divergence of giant viruses predated the divergence of eukaryotes (line 200). In that case, the host analyses should also include prokaryotes instead of assuming these viruses only infect eukaryotes.

Thank you for your thought-provoking comment. Indeed, it is currently believed that all viruses of the phylum Nucleocytoviricota infect eukaryotes. While some members of the Duplodnavirus group (which includes the Mirusviruses) are known to infect bacteria (as bacteriophages), multiple pieces of evidence support the notion that Mirusviruses primarily infect eukaryotes. These evidences include horizontal gene transfer events, size fraction of assembly library data, metatranscriptome, cooccurrence/correlation analyses and the presence of TATA-binding proteins. Therefore, we believe that our assumption that all viruses in the GOEV database infect eukaryotes is warranted.

We acknowledge that the earliest members of the giant viral lineage may have infected organisms that predate the divergence of eukaryotes (LECA). However, since the focus of this manuscript is on the ecological patterns rather than the deep evolutionary history of the Nucleocytoviricota/Mirusviricota, we consider this topic to be beyond the scope of our current analysis. We appreciate your valuable insights.

Minor comments:

1. line 178: Here *Emiliana huxleyi* is used as an example of polar-distributed species, but it seems that it is cosmopolitan in its distribution (including tropical regions). Also many of their viruses seem to have been isolated from nonpolar regions.

Upon further review, we agree that *Emiliana huxleyi* is not a suitable example of a polar-distributed species, given its cosmopolitan distribution. We have accordingly revised our manuscript to remove this reference and used:

(Ln 195-196)---*“For example, Chrysochromulina ericina virus, isolated from high latitude Norwegian coastal waters, was correctly assigned to the Polar niche.”*---

2. line 70: Ref 27 doesn't appear to say much about the overall dynamics of losing and gaining genes in giant viruses. Probably some other studies focused on gene repertoire evolution should be included here.

Thank you for your comment. Actually in the reference 27, Schulz et al, 2017,

<https://pubmed.ncbi.nlm.nih.gov/28386012/> write :

“We observed gene gain, exceeding the amount of gene loss and leading to substantial genome size increase, in each of these three lineages independently. Lineage-specific gene gain is particularly prevalent in three of the four Klosneuviruses (Fig. 2), resulting in divergent gene complements (Figs. 1 and 2). The reconstructed viral genome evolution follows the accordion model in which phases of preferential gene gain and gene family expansion alternate with gene-loss dominated phases (18) (Fig. 2). Differences and convergence among viral lineages are likely due to adaptation of different giant viruses to distinct hosts after host switches.”

In particular, we believe that the right panel of Figure 2 of Schulz et al 2017 is of particular relevance to our question, as it addresses gain and loss events grouped by gene function as annotated using nucleocytoplasmic virus orthologous groups (NCVOGs).

To more comprehensively describe these dynamics, we have now added another reference that provide further insight into gene gain and loss in giant viruses. The additional reference allows us to present a more detailed and nuanced understanding of the evolution of giant virus gene repertoires. We thank you for bringing this to our attention.

Koonin et.al. 2019, <https://pubmed.ncbi.nlm.nih.gov/30635076/>

3. line 200: Does Ref 23 provide any evidence to show that the divergence of giant viruses predates the divergence of eukaryotes?

The estimates of relative divergence times for giant viruses and eukaryotes are presented in the other citation (21, Guglielmini et al.), which builds upon the biodiversity estimates presented in Mihara et al. (Ref. 23). We believe these two papers should give the reader a reasonable understanding of how this estimate was made.

4. line 285: Like *Emiliana huxleyi*, *Cafeteria roenbergensis* also seems to have a widespread distribution instead of being polar.

Thank you for bringing this to our attention. You're right in pointing out that *Cafeteria roenbergensis* also appears to have a wide distribution and is not exclusive to polar

regions. In the context of the paragraph, we used Cafeteria roenbergensis virus to illustrate the capability of viral genome to encode a full set of CMP-KDO. However, we appreciate the potential for misunderstanding and have clarified this point in our manuscript by rearranged the words.

(Ln 312-316)---“Enzymes of CMP-KDO biosynthesis were previously found encoded by a giant virus and suggested to add glycoconjugates to the surface of virions to enhance virion-cell recognition⁶¹. The genomes in the examined Polar clade may also coat virions with glycoconjugates, to potentially enhance interactions with their polar hosts and/or virion stability.”---

We hope this neutral way of expression is not misleading now.

5. line 315-316: Is there any proof of concept that the method can really detect polar-adapted genes in eukaryotes?

To be honest the answer to your question is “no”. Our study is based on a simple principal hypothesis that “if a gene is specific to a habitat, then the gene is likely to be important for the organism to survive in the habitat”.

Consequently, our standpoint is that polar adaptive genes (if exist) will be detected as polar specific genes. However, we cannot assert that all polar specific genes are polar-adaptive genes.

Our strategy is to identify gene repertoire with statistically robust geographic distributions, either polar or non-polar. In our study, we used the same method to delineate polar specific genes for both viruses and eukaryotes with the intention to compare their similarities (if they shared similar genomic strategies) and proportions in their genomes to assess the impact of such polar specific genes on the formation of gene repertoire (viruses 20% vs eukaryote 4%).

Within this logical frame, we hypothesized that

(Ln 367-368)--- “viruses appear to take their own strategies that are distinct from the host strategies for their adaption in the same habitat.”

This might only represent partial aspect of the actual polar adaptive processes for viruses and eukaryotes, but our study underscores the need for further exploration of viral adaptation strategies to the polar, as well as other environments.

6. line 321-323: "The result further suggests that virus-host horizontal gene transfer is not the primary driver of viral polar adaptation, and that genomic adaptations are uncoupled between viruses and eukaryotes." If it is not through horizontal gene transfer or during infection of hosts, how could these viruses have obtained these genes in the first place?

We do not discount the role of HGT in the acquisition of viral polar-adapted genes. Indeed, we believe HGT from hosts (or other sources such as viruses) significantly contributes to the giant viruses' repertoire of polar-adapted genes. However, we want to emphasize that the homologs in eukaryotes are not polar-specific. Host homologs for viral polar-adapted genes (that viruses may gain through HGT) may not necessarily contribute to the host's adaptation to polar environments.

This argument is expressed in the last part of discussion:

Ln 386-388: ---"*Numerous functions, especially ones related to host interactions, were found to be specific to viral polar adaptation, but most of them were not identified as polar-specific functions in eukaryotes, suggesting a decoupling of viral and host polar adaptations.*"---

We want to thank the reviewer again for the time and thoughtful comments. Their comments, especially those concerning the sources of viral genes (HGTs), were instrumental in refining our thinking and discussion. We believe that their input has significantly enhanced the logical of our study in the current revision.

REVIEWERS' COMMENTS

Reviewer #1 (Remarks to the Author):

The authors have adequately addressed the points I raised. I do not have any further concerns.

Reviewer #2 (Remarks to the Author):

I'd like to thank the authors for addressing all of my comments. The manuscript is much improved, and it will be a valuable addition to the literature. I'm glad my comments on the EVEs were useful. I appreciate the new details on how the KEGG enrichment was done- sometimes these results can be difficult to interpret, but the new description of the methods shows that it is robust and reliable. One last small detail- I believe some of the references are duplicated.

Best,
Frank Aylward

Reviewer #3 (Remarks to the Author):

The authors have addressed all my comments in the review report. I have no further comments to add.

Reviewer #1 (Remarks to the Author):

The authors have adequately addressed the points I raised. I do not have any further concerns.

We want to thank the reviewer again for helping clarify the concept of viral adaptation and virus-host system. This comment substantially improved the interpretability and readability of the manuscript.

Reviewer #2 (Remarks to the Author):

I'd like to thank the authors for addressing all of my comments. The manuscript is much improved, and it will be a valuable addition to the literature. I'm glad my comments on the EVEs were useful. I appreciate the new details on how the KEGG enrichment was done- sometimes these results can be difficult to interpret, but the new description of the methods shows that it is robust and reliable.

One last small detail- I believe some of the references are duplicated.

Best,

Frank Aylward

We are grateful for the insightful feedback from Prof. Aylward. His comments have greatly enhanced the standardization of our metagenomic data usage and the overall writing of this paper. We apologize for the oversight regarding the duplicated reference (Irwin, N. A. T., Pittis, A. A., Richards, T. A. & Keeling, P. J. Systematic evaluation of horizontal gene transfer between eukaryotes and viruses. *Nat Microbiol* 7, 327–336 (2022).; Moniruzzaman, M., Weinheimer, A. R., Martinez-Gutierrez, C. A. & Aylward, F. O. Widespread endogenization of giant viruses shapes genomes of green algae. *Nature* 588, 141–145 (2020).), we have corrected the list.

Reviewer #3 (Remarks to the Author):

The authors have addressed all my comments in the review report. I have no further comments to add.

We want to thank the reviewer again for the time and thoughtful comments. Their comments, especially those concerning the sources of viral genes (HGTs), have significantly enhanced the quality of our manuscript.